# Activation of MAP3K DLK and LZK in Purkinje cells causes rapid and slow degeneration depending on signaling strength

Yunbo Li[1], Erin M Ritchie[1], Christopher L Steinke[1], Cai Qi[1†], Lizhen Chen[1‡], Binhai Zheng[2,3], Yishi Jin[1,2,4]*

[1]Neurobiology Section, Division of Biological Sciences, University of California San Diego, La Jolla, United States; [2]Department of Neurosciences, School of Medicine, University of California San Diego, La Jolla, United States; [3]VA San Diego Healthcare System, San Diego, United States; [4]Kavli Institute of Brain and Mind, University of California San Diego, La Jolla, United States

*For correspondence:
yijin@ucsd.edu

Present address: † Department of Human Genetics, and the Grossman Institute for Neuroscience, Quantitative Biology and Human Behavior, The University of Chicago, Chicago, United States; ‡ Barshop Institute for Longevity and Aging Studies, Department of Cell Systems and Anatomy, University of Texas Health Science Center at San Antonio, San Antonio, United States

Competing interests: The authors declare that no competing interests exist.

**Abstract** The conserved MAP3K Dual-Leucine-Zipper Kinase (DLK) and Leucine-Zipper-bearing Kinase (LZK) can activate JNK via MKK4 or MKK7. These two MAP3Ks share similar biochemical activities and undergo auto-activation upon increased expression. Depending on cell-type and nature of insults DLK and LZK can induce pro-regenerative, pro-apoptotic or pro-degenerative responses, although the mechanistic basis of their action is not well understood. Here, we investigated these two MAP3Ks in cerebellar Purkinje cells using loss- and gain-of function mouse models. While loss of each or both kinases does not cause discernible defects in Purkinje cells, activating DLK causes rapid death and activating LZK leads to slow degeneration. Each kinase induces JNK activation and caspase-mediated apoptosis independent of each other. Significantly, deleting CELF2, which regulates alternative splicing of *Map2k7*, strongly attenuates Purkinje cell degeneration induced by LZK, but not DLK. Thus, controlling the activity levels of DLK and LZK is critical for neuronal survival and health.

## Introduction

Mitogen-activated protein kinase (MAPK) signaling pathways play important roles in neuronal development and function, and aberrant regulation of MAP kinases is associated with many neurological diseases, such as Parkinson's disease (PD), amyotrophic lateral sclerosis (ALS) and Alzheimer's disease (AD) (*Thomas and Huganir, 2004*; *Schellino et al., 2019*; *Hotamisligil and Davis, 2016*; *Hollville et al., 2019*; *Asghari Adib et al., 2018*). The MAPK cascade involves MAP3Ks (MAP kinase kinase kinases), MAP2Ks and MAPKs that together form a phosphorylation relay and activate downstream signaling events in response to external or internal stimuli. The mammalian MAP3K DLK (Dual leucine zipper kinase, or MAP3K12) and LZK (Leucine zipper kinase, or MAP3K13) are orthologs of *C. elegans* DLK-1 and *Drosophila* DLK/Wallenda (*Jin and Zheng, 2019*). These MAP3Ks act as upstream kinases for JNK and p38 MAP kinase, and are now known as key players in neuronal stress response network both under acute injury and in chronic neurodegenerative diseases (*Tedeschi and Bradke, 2013*; *Jin and Zheng, 2019*; *Asghari Adib et al., 2018*; *Farley and Watkins, 2018*). An emerging theme is that while activation of these kinases triggers seemingly common pathways, the outcome is highly context-specific both in terms of cell types and forms of insults.

Both DLK and LZK show broad expression in the nervous system. Several studies have investigated roles of DLK in the development of the nervous system. Constitutive DLK knockout mice die

perinatally, and different regions of developing brain display varying degrees of altered axon fibers, abnormal synapses and increased neuronal survival (*Hirai et al., 2006*; *Hirai et al., 2011*; *Itoh et al., 2011*; *Sengupta Ghosh et al., 2011*). However, mice with adult deletion of DLK survive and show no detectable abnormalities (*Pozniak et al., 2013*; *Le Pichon et al., 2017*). Under traumatic insults, DLK activity is reported to increase and trigger a variety of cellular responses. For example, sciatic nerve injury induces DLK-dependent pro-regenerative responses in dorsal root ganglia (DRG) sensory neurons (*Shin et al., 2012*; *Shin et al., 2019*). In the CNS, optic nerve injury upregulates DLK expression in retinal ganglion cells (RGC), which triggers cell death and also promotes axon growth from surviving RGCs (*Watkins et al., 2013*; *Welsbie et al., 2013*). In a mouse model for stroke, increased DLK expression in pre-motor cortex is suggested to promote motor recovery (*Joy et al., 2019*). Increased DLK activity is also reported in animal models of neurodegeneration, and genetically or pharmacologically inhibiting DLK in the aged PS2APP mice for AD and the SOD1$^{G93A}$ mice for ALS has resulted in some neuroprotective effects (*Le Pichon et al., 2017*). Additionally, in human iPSC derived neurons treated with ApoE4, a protein associated with an increased risk for AD, DLK is rapidly upregulated and enhances transcription of APP (*Huang et al., 2017*). As numerous approaches now target DLK for drug discovery (*Siu et al., 2018*), it is important to investigate how the pleiotropic effects of manipulating DLK in different cell types influence disease progression. In comparison, despite the fact that LZK was also discovered 20 years ago (*Sakuma et al., 1997*), the in vivo roles of LZK are only beginning to be explored. In a mouse model of spinal cord injury, LZK is upregulated in astrocytes and mediates reactive astrogliosis (*Chen et al., 2018*). Emerging studies show that LZK can cooperate with DLK to promote RGC death after optic nerve injury and DRG axon degeneration (*Welsbie et al., 2017*; *Summers et al., 2020*).

Here, we dissect the roles of these two kinases in the cerebellar Purkinje cells. We analyzed genetic deletion mice for each kinase, and also developed transgenic mice that allow for Cre-mediated expression of DLK or LZK. Biochemical studies have shown that DLK and LZK undergo auto-activation via leucine-zipper-mediated dimerization, and such auto-activation is dependent on the protein abundance (*Nihalani et al., 2000*; *Ikeda et al., 2001b*). Therefore, elevating expression of DLK or LZK is a proxy to its activation of the downstream signal transduction. We find that deletion of DLK and/or LZK, singly or in combination, from Purkinje cells, does not affect their development and postnatal growth. In contrast, induced expression of DLK in Purkinje cells causes rapid degeneration, whereas elevating LZK expression induces slow degeneration. Strikingly, we find that deleting the RNA splicing factor CELF2 ameliorates Purkinje cell degeneration induced by LZK, but not DLK, partly via regulating alternative splicing of *Map2k7*, which encodes the MAP2K7/MKK7. These findings provide important insights to the understanding of neurodegenerative processes.

## Results

### Normal development of cerebellar Purkinje cells in the absence of DLK and LZK

Both DLK and LZK are expressed in cerebellar neurons, with high levels of DLK observed in the molecular layer of adult cerebellum (*Hirai et al., 2005*; *Suenaga et al., 2006*; *Goodwani et al., 2020*). DLK knockout (KO) mice die soon after birth, and cerebellar architecture is grossly normal (*Hirai et al., 2006*). The roles of LZK in neuronal development remain unknown. To address the function of LZK and further probe into the interactions between the two kinases, we generated an LZK KO mouse line (genotype *Map3k13$^{KO/KO}$*) using CRISPR-editing to delete the entire kinase domain (*Figure 1—figure supplement 1A–B*), which also resulted in a frameshift and produced no detectable LZK proteins by immunoprecipitation and western blotting analysis (*Figure 1—figure supplement 1C*). The LZK KO mice are viable, and grow indistinguishably from littermates under standard housing conditions. Histological analysis of cerebellar tissue sections using hematoxylin and eosin staining revealed no discernible defects in overall cellular architecture in 2-month-old (P60) LZK KO mice (*Figure 1A*). Immunostaining using antibodies to Calbindin, which specifically labels Purkinje cells, showed that the position, number and gross morphology of Purkinje cells were comparable between LZK KO and control (*Figure 1A–C*). The molecular layer thickness, which is a sensitive assessment for disruption of the dendrites of Purkinje cells (*White et al., 2014*; *Hansen et al., 2013*), was also normal (*Figure 1D*).

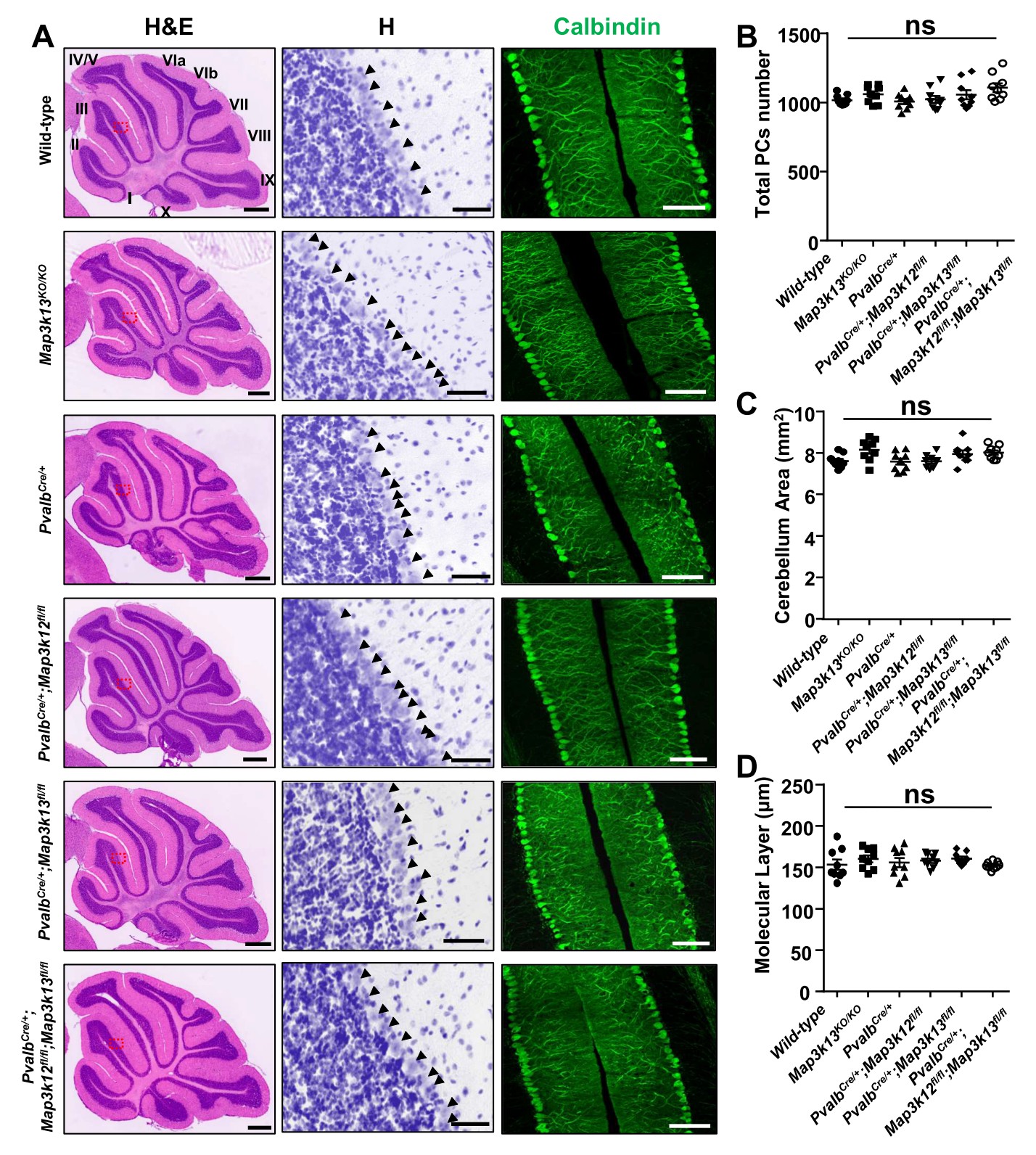

**Figure 1.** Morphology of cerebellar Purkinje cells is normal in the absence of LZK and DLK. (**A**) Histological and immunostaining images of cerebellar sections of P60 mice of genotypes indicated, stained with hematoxylin and eosin (left panel), hematoxylin (enlarged red boxed areas, middle panel) and Calbindin (right panel). *Map3k12* synonyms for DLK, *Map3k13* for LZK. Roman letters mark corresponding lobules in the vermis of the cerebellum. Arrows point to the Purkinje cell bodies. Scale bars: 500 μm (left panel), 50 μm (middle panel), 100 μm (right panel). (**B**) Quantification of total number of

*Figure 1 continued on next page*

*Figure 1 continued*

Purkinje cells in all cerebellar lobules. (**C**) Quantification of cerebellum area, with perimeters measured by outlining the outer edge of midline sagittal sections of the cerebella. (**D**) Quantification of the molecular layer thickness in cerebellar lobules V-VI. (**B–D**) n = 3 animals per genotype, and three sections/animal; data shown as means ± SEM. Statistics: one-way ANOVA; ns. no significant.

The online version of this article includes the following source data and figure supplement(s) for figure 1:

**Source data 1.** Total number of Purkinje cells in all cerebellar lobules of P60 mice of genotypes indicated.
**Source data 2.** Cerebellum area of P60 mice of genotypes indicated.
**Source data 3.** Molecular layer thickness in cerebellar lobules V-VI of P60 mice of genotypes indicated.
**Figure supplement 1.** Knockout mice lines and associated evidence.

It has recently been shown that DLK and LZK can act synergistically in injured RGC or DRG neurons (*Welsbie et al., 2017*; *Summers et al., 2020*). We therefore tested if loss of both DLK and LZK might affect cerebellar neurons. We bred floxed (*fl*) KO mice for DLK (genotype *Map3k12^{fl/fl}*) or LZK (genotype *Map3k13^{fl/fl}*) to a parvalbumin-Cre driver line (*Pvalb^{Cre}*) (*Hippenmeyer et al., 2005*). We obtained *Pvalb^{Cre/+};Map3k12^{fl/fl};Map3k13^{fl/fl}* mice, along with *Pvalb^{Cre/+};Map3k12^{fl/fl}* and *Pvalb^{Cre/+}; Map3k13^{fl/fl}* mice (*Figure 1—figure supplement 1D–G*). Cre recombinase from *Pvalb^{Cre}* line is active in Purkinje cells as early as P4 (*Hippenmeyer et al., 2005*). We detected reduced protein levels of DLK and LZK in cerebellar extracts for *Pvalb^{Cre/+};Map3k12^{fl/fl};Map3k13^{fl/fl}*, *Pvalb^{Cre/+};Map3k12^{fl/fl}* and *Pvalb^{Cre/+};Map3k13^{fl/fl}* mice, respectively (*Figure 1—figure supplement 1C,H*). The overall cerebellar tissue organization, revealed by hematoxylin and eosin staining, was indistinguishable among test and control mice of P60 age (*Figure 1A*). Calbindin immunostaining showed that the total number of Purkinje cells and the molecular layer thickness of cerebellum were comparable in single or double deletion of each kinase gene (*Figure 1A–D*). Additionally, GFAP immunostaining for cerebellar astrocytes revealed no detectable difference among different genotypes of mice (*Figure 1—figure supplement 1I–J*). All mutant mice also showed normal postnatal growth, measured by body weight, comparable to the control mice under same housing conditions (*Figure 1—figure supplement 1K*). These data show that DLK and LZK are not required for the postnatal development of Purkinje cells.

## Elevating DLK expression in Purkinje cells causes rapid degeneration via apoptosis

Increased expression of DLK or LZK has been observed under traumatic injury or other stress conditions (*Shin et al., 2012*; *Shin et al., 2019*; *Watkins et al., 2013*; *Welsbie et al., 2013*; *Joy et al., 2019*; *Chen et al., 2018*; *Huang et al., 2017*). We next investigated how elevating expression of DLK and LZK, which would lead to activation of these kinases, affects neurons. To this end, we generated two transgenic mouse lines by inserting a Cre-inducible DLK or LZK cDNA expression construct at the *Hipp11* locus, designated as Hipp11-DLK(iOE) or Hipp11-LZK(iOE), respectively (*Figure 2—figure supplement 1A–B*). In each transgene, the induced expression can be readily assessed by a tdTomato reporter fused in-frame to the C-terminus of DLK or LZK through the T2A self-cleaving peptides. By RNA-seq analysis we detected comparable levels of tdTomato mRNAs produced from each transgene following expression of Cre recombinase (*Figure 2—figure supplement 1C*). After outcrossing to C57BL/6J background, these mice were bred to the *Pvalb^{Cre}* line. In *Pvalb^{Cre/+};Hipp11-DLK(iOE)/+* or *Pvalb^{Cre/+};Hipp11-LZK(iOE)/+* heterozygous mice, we observed tdTomato reporter expression correlating with the timing of parvalbumin expression (*Figure 2—figure supplement 1D*), and increased DLK or LZK expression was detected in cerebellar protein lysates (*Figure 2—figure supplement 1E–H*).

Elevated DLK expression in *Pvalb^{Cre/+};Hipp11-DLK(iOE)/+* mice caused gross abnormalities noticeable as early as P6. The pups were smaller than littermate controls (*Figure 2A–B*), and exhibited abnormal movements (*Video 1*). These pups all died around P21, with cerebella smaller than those of littermate controls (*Figure 2C*; *Figure 2—figure supplement 2A*). Histological analysis of cerebellar tissues revealed grossly abnormal lobular morphology (*Figure 2—figure supplement 2B*). By Calbindin immunostaining, we detected dramatic degeneration of Purkinje cells, with a nearly

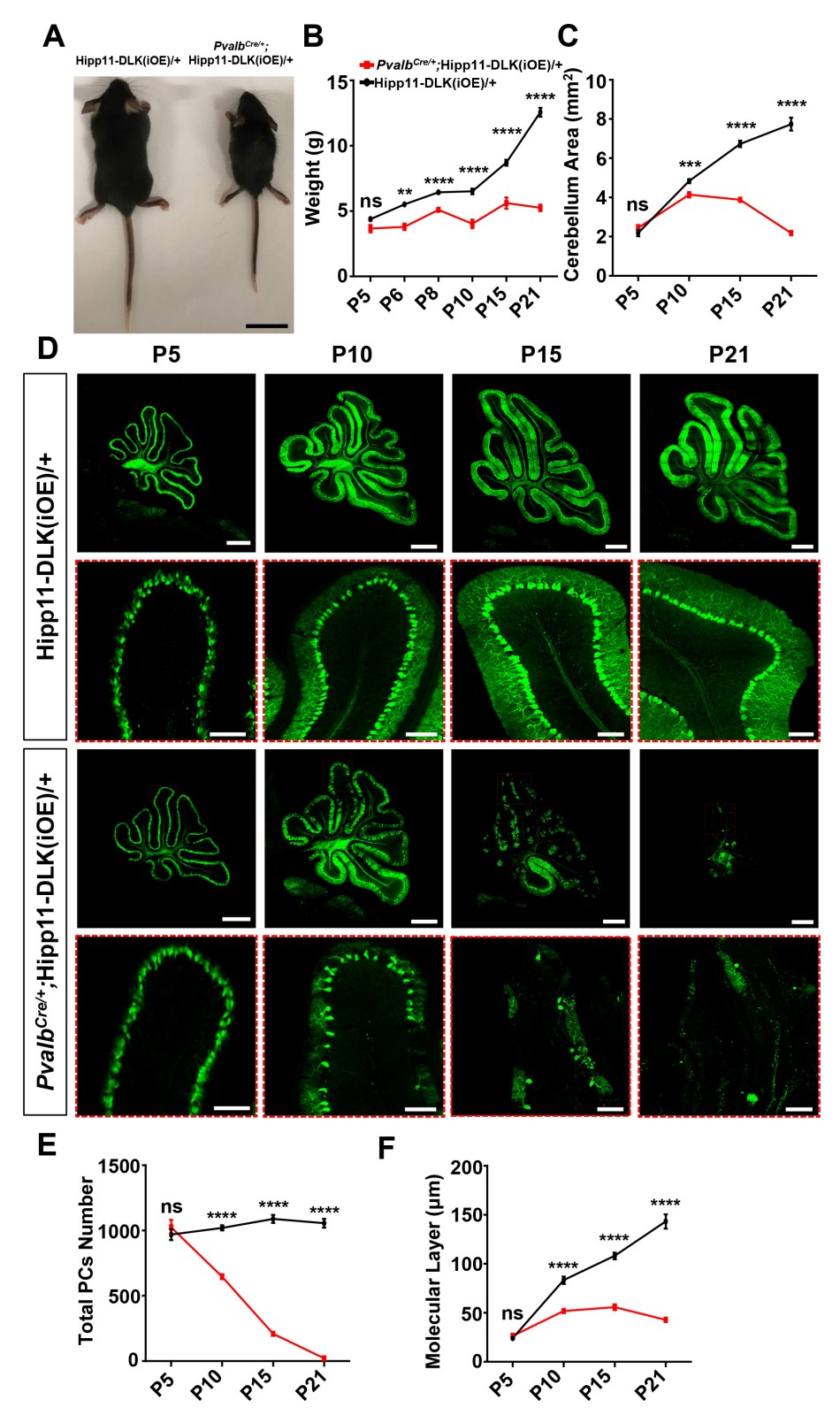

**Figure 2.** Induced expression of DLK using parvalbumin-cre (*Pvalb*$^{Cre/+}$) causes animal growth defects and rapid degeneration of Purkinje cells. (**A**) Representative image of P21 pups of genotypes indicated, and pups with induced DLK expression in PV$^+$ neurons are smaller than siblings. Scale bar: 2 cm. (**B**) Quantification of the body weight from P5 to P21. Hipp11-DLK(iOE)/+: n = 3 (P5), n = 3 (P6), n = 4 (P8), n = 12 (P10), n = 8 (P15) and n = 10 (P21); *Pvalb*$^{Cre/+}$;Hipp11-DLK(iOE)/+: n = 5 (P5), n = 6 (P6), n = 4 (P8), n = 11 (P10), n = 9 (P15) and n = 10 (P21). (**C**) Quantification of the cerebellum area from

*Figure 2 continued on next page*

*Figure 2 continued*

P5 to P21. (D) Representative images of Calbindin staining of cerebellar sections of littermates from mating parents *Pvalb*<sup>Cre/+</sup> to Hipp11-DLK(iOE)/+ at indicated postnatal days. Red boxes are enlarged to show that induced expression of DLK causes a total loss of Purkinje cells by P21. Scale bars: 500 μm (upper panels), 100 μm (lower panels). (E) Quantification of total number of Purkinje cells in all cerebellar lobules. (F) Quantification of the molecular layer thickness in cerebellar lobules V-VI. Color representation for genotypes in C, E, and F is the same as in B. (C, E, F) n = 3 animals per genotype, three sections/animal for each time point. (B–C, E–F) Data shown as means ± SEM. Statistics: Student's unpaired t-test; ns, no significant; **, p<0.01; ***, p<0.001; ****, p<0.0001.

The online version of this article includes the following source data and figure supplement(s) for figure 2:

**Source data 1.** Mouse body weight of genotypes indicated from P5 to P21.
**Source data 2.** Mouse cerebellum area of genotypes indicated from P5 to P21.
**Source data 3.** Total number of Purkinje cells in all cerebellar lobules of mice genotypes indicated from P5 to P21.
**Source data 4.** Molecular layer thickness in cerebellar lobules V-VI of mice genotypes indicated from P5 to P21.
**Figure supplement 1.** Cre-dependent DLK and LZK expression transgenic mice and associated evidence.
**Figure supplement 2.** Additional evidence of Purkinje cell degeneration phenotypes caused by elevated DLK expression.

complete (~98%) cell loss by P21 (*Figure 2D–E*). The molecular layer of the cerebellum in these mice was significantly thinner than that in the littermate control mice from P10 to P21 (*Figure 2F*). Purkinje cell degeneration is known to be associated with increased reactivity of astrocytes and microglia (*Cvetanovic et al., 2015*; *Lobsiger and Cleveland, 2007*; *Lattke et al., 2017*). Indeed, we observed increased expression of GFAP and IBA1 (detecting both microglia and macrophage) in these mice at P21, compared to control mice (*Figure 2—figure supplement 2C–F*). Some IBA1-positive microglia were found to be closely associated with tdTomato-labeled Purkinje cells (*Figure 2— figure supplement 2C*), suggesting that dying cells might be phagocytosed.

To determine if targeted expression of DLK induced activation of the JNK signaling, we co-immunostained for phospho-c-Jun (p-c-Jun) and Calbindin on cerebellar tissue sections of P6 mice. While many p-c-Jun signals were likely from granule neurons as they did not overlap with Calbindin<sup>+</sup> Purkinje cells in both mutant and control mice, we observed that elevated DLK expression induced substantially increased p-c-Jun in Purkinje cells of *Pvalb*<sup>Cre/+</sup>;Hipp11-DLK(iOE)/+ mice, compared to littermate controls (*Figure 3A–B*; *Figure 3—figure supplement 1*). We also asked if the loss of Purkinje cells involved apoptosis using the TUNEL assay. During early postnatal cerebellar development, multiple types of cells undergo apoptosis, including those in the granular and the molecular layers of the cerebellar cortex (*Cheng et al., 2011*). Indeed, in the control littermates, we observed many TUNEL signals at P5, which decreased over the following postnatal days (*Figure 3—figure supplement 2A–B*). In the *Pvalb*<sup>Cre/+</sup>;Hipp11-DLK(iOE)/+ mice, the number of apoptotic cells at P5 was comparable to that in control, but continued to rise over the next 10 days, reaching peak levels around P15 (*Figure 3—figure supplement 2A–B*). Importantly, some TUNEL signals in P15 *Pvalb*<sup>Cre/+</sup>;Hipp11-DLK(iOE)/+ mice co-localized with tdTomato-labeled Purkinje cells (*Figure 3— figure supplement 2C*). Furthermore, a significant portion of the Purkinje cells in the P15 *Pvalb*<sup>Cre/+</sup>;Hipp11-DLK(iOE)/+ mice were positively stained for cleaved caspase-3 (*Figure 3C–D*), a marker for activating apoptosis (*Elmore, 2007*). These results show that elevating DLK expression in Purkinje cells activates the JNK pathway and causes early-onset, rapid degeneration through apoptotic cell death.

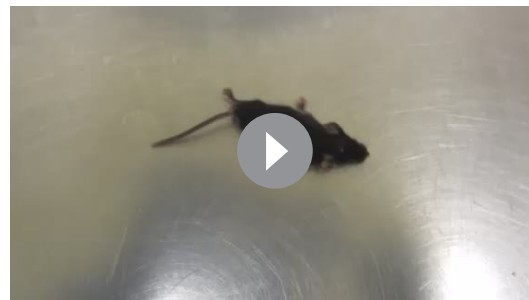

**Video 1.** Locomotor deficits in P15 *Pvalb*<sup>Cre/+</sup>;Hipp11-DLK(iOE)/+ mice. A representative P15 old *Pvalb*<sup>Cre/+</sup>;Hipp11-DLK(iOE)/+ mouse exhibited difficulty moving forward and drags its abdomen along the ground, as well as tremors.
https://elifesciences.org/articles/63509#video1

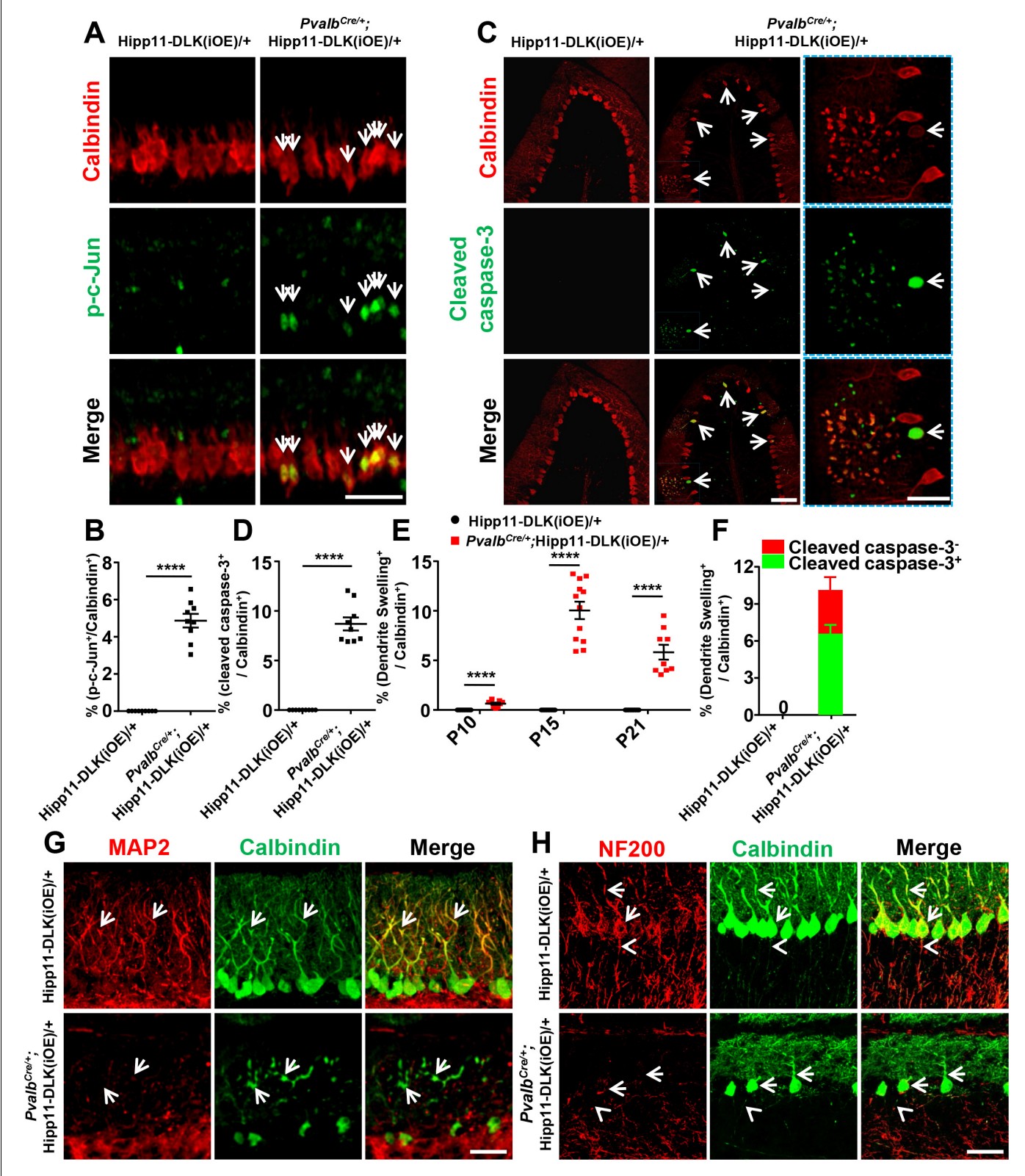

**Figure 3.** Induced expression of DLK activates JNK signaling and apoptosis, and disrupts dendritic cytoskeleton of Purkinje cells. (**A**) Representative images of co-immunostaining of p-c-Jun and Calbindin in Purkinje cells in P6 mice of genotypes indicated. Arrows point to p-c-Jun immunostaining signals in nuclei of Purkinje cells. Scale bar: 100 μm. (**B**) Quantification of percentage of p-c-Jun[+] signals per total Purkinje cells in P6 mice of Hipp11-DLK(iOE)/+ and *Pvalb*[Cre/+];Hipp11-DLK(iOE)/+. (**C**) Representative images of cerebellar lobule III of P15 mice, co-immunostained for Calbindin and

*Figure 3 continued on next page*

*Figure 3 continued*

cleaved caspase-3, with arrows and enlarged blue boxes showing cleaved caspase-3$^+$ signals in the swollen dendrites of Purkinje cells of *Pvalb$^{Cre/+}$*; Hipp11-DLK(iOE)/+ mice. Scale bars: 100 μm (left and middle panels), 50 μm (right panel). (D) Quantification of percentage of cleaved caspase-3$^+$ cells in total Purkinje cells in P15 mice of Hipp11-DLK(iOE)/+ and *Pvalb$^{Cre/+}$*;Hipp11-DLK(iOE)/+, and none detected in Hipp11-DLK(iOE)/+. (E) Quantification of percentage of Purkinje cells with swelling dendrites in mice of Hipp11-DLK(iOE)/+ and *Pvalb$^{Cre/+}$*;Hipp11-DLK(iOE)/+ from P10 to P21, and none detected in Hipp11-DLK(iOE)/+. Hipp11-DLK(iOE)/+: n = 3 animals (P10, P15, P21), three sections/animal; *Pvalb$^{Cre/+}$*;Hipp11-DLK(iOE)/+: n = 3 (P10, P21), n = 4 (P15) animals, three sections/animal. (F) Quantification of percentage of Purkinje cells containing cleaved caspase-3$^+$ in swelling dendrites in total Purkinje cells in P15 mice of Hipp11-DLK(iOE)/+ and *Pvalb$^{Cre/+}$*;Hipp11-DLK(iOE)/+. (G) Representative images of Purkinje cells of P15 mice co-immunostained for MAP2 and Calbindin. Swelling dendrites in Purkinje cells of *Pvalb$^{Cre/+}$*;Hipp11-DLK(iOE)/+ mice have little expression of MAP2; arrows point to dendrites with intensity difference between the two genotypes. Scale bar: 50 μm. (H) Representative images of Purkinje cells of P15 mice co-immunostained for NF-200 and Calbindin, showing that induced DLK expression reduces NF-200 staining in dendrites (arrows) and axons (arrowheads) of Purkinje cells. Scale bar: 50 μm. (B, D, F) n = 3 animals per genotype, three sections/animal. (B, D, E, F) Data shown are means ± SEM. Statistics: Student's unpaired t-test; ****, p<0.0001.

The online version of this article includes the following source data and figure supplement(s) for figure 3:

**Source data 1.** Percentage of p-c-Jun$^+$ Purkinje cells in total Purkinje cells in P6 mice of genotypes indicated.

**Source data 2.** Percentage of cleaved caspase-3$^+$ Purkinje cells in total Purkinje cells in P15 mice of genotypes indicated.

**Source data 3.** Percentage of Purkinje cells with swelling dendrites in mice of genotypes indicated from P10 to P21.

**Source data 4.** Percentage of Purkinje cells containing cleaved caspase-3$^+$ in swelling dendrites in total Purkinje cells in P15 mice of genotypes indicated.

**Figure supplement 1.** Additional evidence for p-c-Jun expression in cortex of *Pvalb$^{Cre/+}$*;Hipp11-DLK(iOE)/+ mice.

**Figure supplement 2.** Additional evidence for apoptosis induced by elevated DLK expression.

**Figure supplement 3.** DLK protein localization in Purkinje cells.

**Figure supplement 4.** Additional images showing Purkinje cells that have dendrite swelling but are cleaved caspase-3$^-$ in *Pvalb$^{Cre/+}$*;Hipp11-DLK(iOE)/+ mice at P15. Scale bar: 50 μm.

## Elevated DLK expression disrupts dendritic cytoskeleton

DLK is known to be localized to neuronal processes (*Suenaga et al., 2006*) and regulates microtubule stability (*Simard-Bisson et al., 2017*; *Valakh et al., 2015*; *Hirai et al., 2011*). By immunostaining using anti-DLK antibodies on cerebellar sections of *Pvalb$^{Cre/+}$*;Hipp11-DLK(iOE)/+ mice, we detected DLK expression in the somas, dendrites and axons of Purkinje cells (*Figure 3—figure supplement 3*). In these mice, a substantial portion of Purkinje cells, visualized by Calbindin, showed numerous varicosities along the dendrite arbors (*Figure 3C,E*, annotated dendrite swelling). Moreover, ~65% of Purkinje cells with dendrite swelling were positively stained for cleaved caspase-3 (*Figure 3F*; *Figure 3—figure supplement 4*). The percentage of Purkinje cells with dendrite swelling was highest around P15 (*Figure 3E*), consistent with the time course of Purkinje cell death caused by increased DLK expression.

Dendrite swelling is associated with major disorganization of the cytoskeleton network (*Cupolillo et al., 2016*; *Liu et al., 2015*; *Hoskison et al., 2007*). We next assessed how the microtubule cytoskeleton was altered by immunostaining for microtubule-associated protein 2 (MAP2), which is expressed in dendrites of Purkinje cells (*Dehmelt and Halpain, 2005*). We found that MAP2 levels were significantly decreased in dendrites of Purkinje cells in *Pvalb$^{Cre/+}$*;Hipp11-DLK(iOE)/+ mice, compared to the levels of Calbindin as well as to control mice at P15 (*Figure 3G*). The neurofilament protein NF-200 is present in both dendrites and axons and implicated in axon growth and regeneration (*Wang et al., 2012*). By immunostaining, we observed decreased levels of NF-200 in axons and dendrites of Purkinje cells in *Pvalb$^{Cre/+}$*;Hipp11-DLK(iOE)/+ mice (*Figure 3H*). These data are consistent with the notion that DLK regulates the neuronal cytoskeleton, and suggest that the dendritic cytoskeleton in Purkinje cells may be prone for disruption upon aberrant activation of DLK, although we do not exclude the possibility that the dendrite cytoskeleton disruption could be a nonspecific downstream effect.

## Elevating LZK expression in Purkinje cells causes late degeneration

In contrast to the early lethality of *Pvalb$^{Cre/+}$*;Hipp11-DLK(iOE)/+ pups, the *Pvalb$^{Cre/+}$*;Hipp11-LZK (iOE)/+ mice survived to older adults (observed up to 8 months). The adult mice had low body weight, compared to control mice (*Figure 4—figure supplement 1A–B*). Histological analysis

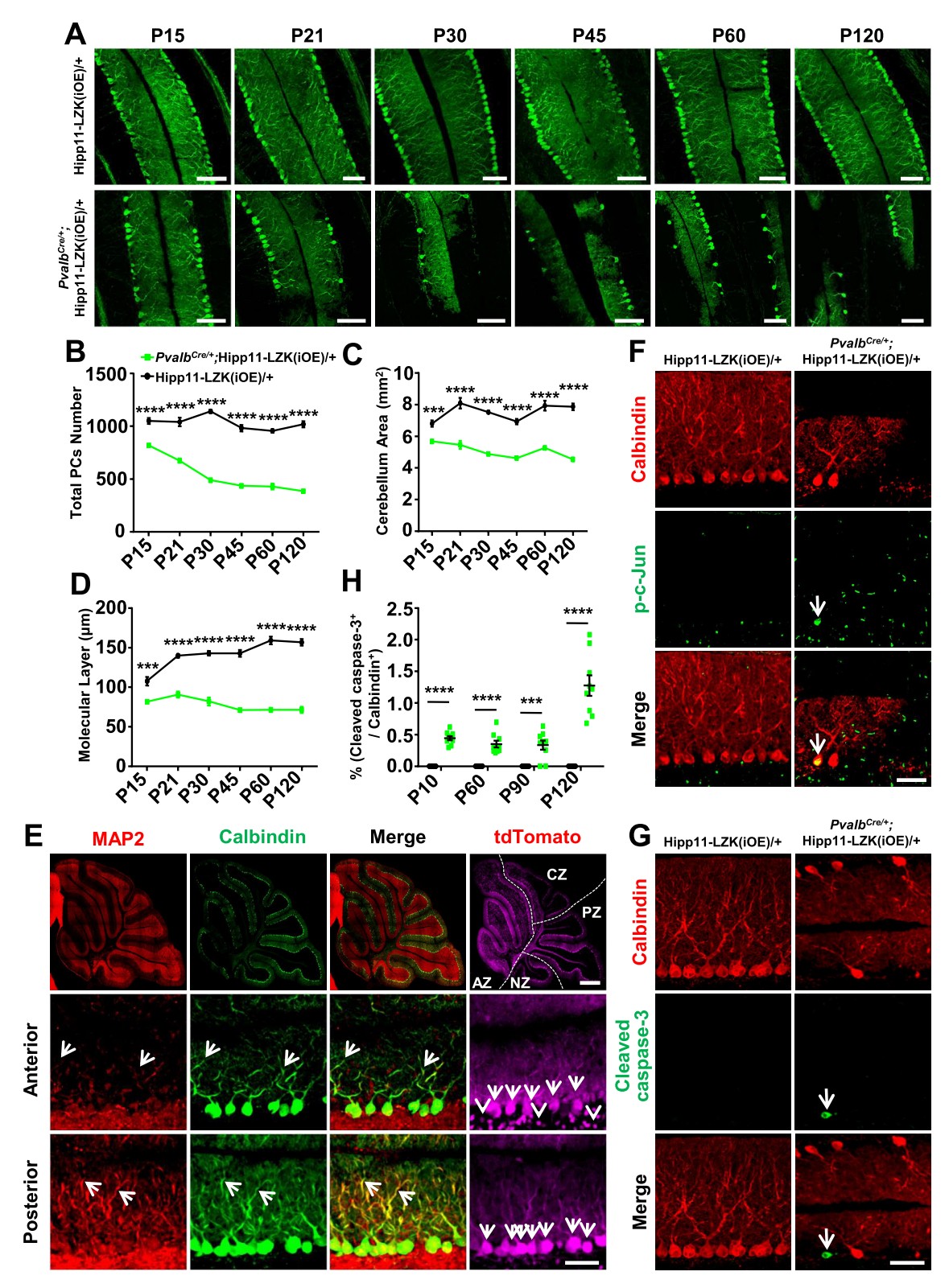

**Figure 4.** Induced expression of LZK using *Pvalb^Cre^* causes progressive degeneration of Purkinje cells. (**A**) Representative images of Calbindin staining of cerebellar sections of littermates from mating parents *Pvalb^Cre/+^* to Hipp11-LZK(iOE)/+ at indicated postnatal days. Scale bars: 100 μm. (**B**) Quantification of total Purkinje cells in all cerebellar lobules. (**C**) Quantification of cerebellum area, perimeters measured by outlining the outer edge of cerebellum. (**D**) Quantification of the molecular layer thickness in cerebellar lobules V-VI. (**E**) Representative images of cerebellar sections of *Pvalb^Cre/+^*;

*Figure 4 continued on next page*

*Figure 4 continued*

Hipp11-LZK(iOE)/+ mice at P15, co-immunostained for MAP2 and Calbindin. The dotted lines in top panels mark the boundary of anterior zone (AZ; lobules I–V), central zone (CZ; lobules VI–VII), posterior zone (PZ; lobules VIII–XI) and nodular zone (NZ; lobule X) in cerebellum, note higher tdTomato intensity in anterior cerebellum (arrows and arrowheads). Images in the bottom two rows show enlarged views of the anterior and posterior cerebellum, with arrows pointing to dendrites of Purkinje cells. Scale bars: 500 µm (upper panel), 100 µm (middle and lower panels). (F) Representative images of Purkinje cells of P90 mice co-immunostained for Calbindin and p-c-Jun. Scale bar: 50 µm. (G) Representative images of Purkinje cells of P90 mice co-immunostained for Calbindin and cleaved caspase-3. Note that *Pvalb*$^{Cre/+}$;Hipp11-LZK(iOE)/+ mice have smaller cerebella, hence two rows of Purkinje cells are in view. Scale bar: 50 µm. (H) Quantification of the percentage of cleaved caspase-3$^+$ Purkinje cells in total Purkinje cells in Hipp11-LZK(iOE)/+ and *Pvalb*$^{Cre/+}$;Hipp11-LZK(iOE)/+ mice from P10 to P120. Color representation for genotypes in C, D, H is the same as in B. (B–D, H) n = 3 animals per genotype, three sections/animal for each time point; data shown are means ± SEM. Statistics: Student's unpaired t-test; ***, p<0.001; ****, p<0.0001. The online version of this article includes the following source data and figure supplement(s) for figure 4:

**Source data 1.** Total number of Purkinje cells in all cerebellar lobules of mice genotypes indicated from P15 to P120.
**Source data 2.** Mouse cerebellum area of genotypes indicated from P15 to P120.
**Source data 3.** Molecular layer thickness in cerebellar lobules V–VI of mice genotypes indicated from P15 to P120.
**Source data 4.** Percentage of cleaved caspase-3$^+$ Purkinje cells in total Purkinje cells in mice genotypes indicated from P10 to P120.
**Figure supplement 1.** Additional evidence for elevating LZK expression in PV$^+$ neurons causing animal growth and movement defects.
**Figure supplement 2.** Additional evidence for degeneration of Purkinje cells induced by elevated LZK expression.
**Figure supplement 3.** Additional evidence of p-c-Jun expression in cortex of *Pvalb*$^{Cre/+}$;Hipp11-LZK(iOE)/+ mice.

showed all lobular structures were present in P120 *Pvalb*$^{Cre/+}$;Hipp11-LZK(iOE)/+ mice (*Figure 4—figure supplement 1C*). By Calbindin immunostaining we observed morphological abnormalities of Purkinje cells around P15, with severity and cell loss increasing from P21 to P120 (*Figure 4A–B*). The area of cerebellum and the molecular layer were also reduced significantly (*Figure 4C–D*). A previous study reported that *Pvalb*$^{Cre}$ can induce higher levels of transgene expression in anterior zone than other zones of cerebellum (*Asrican et al., 2013*). Consistently, we observed that the fluorescence intensities of tdTomato and LZK immunostaining were stronger in the anterior lobules than those in posterior cerebellum (*Figure 4E*; *Figure 4—figure supplement 1D–E*). Correlating with the expression levels of LZK, there were more Purkinje cells with severe abnormal cell morphology in the anterior cerebellum than the posterior (*Figure 4E*). Reduced MAP2 levels were also more noticeable in Purkinje cells located in the anterior than those in the posterior cerebellum (*Figure 4E*). Additionally, GFAP staining showed significantly increased astrocyte reactivity in *Pvalb*$^{Cre/+}$;Hipp11-LZK(iOE)/+ mice from P21 (*Figure 4—figure supplement 2A–B*), particularly in areas surrounding Purkinje cells (*Figure 4—figure supplement 2C*).

We addressed whether Purkinje cell degeneration caused by LZK involved JNK activation and induction of apoptosis. While no detectable p-c-Jun was observed in Purkinje cells of P90 control mice, elevating LZK expression increased p-c-Jun in Purkinje cells (*Figure 4F*). Cleaved caspase-3 immunoreactivity co-localized with Purkinje cells in *Pvalb*$^{Cre/+}$;Hipp11-LZK(iOE)/+ mice, but not in control mice (*Figure 4G–H*). Thus, elevating LZK expression also triggers JNK activation and caspase mediated apoptosis in Purkinje cells; however, these cells undergo a slow degeneration process. These data suggest differential regulation of the signaling network induced by elevated expression of DLK and LZK.

## Purkinje cell degeneration induced by LZK overexpression is attenuated by loss of CELF2, a regulator of *Map2k7* alternative splicing

Biochemical studies have shown that two MAP2K, MKK4 and MKK7 act downstream of DLK and LZK to activate JNK (*Hirai et al., 2011*; *Le Pichon et al., 2017*; *Huang et al., 2017*; *Ikeda et al., 2001a*; *Chen et al., 2016b*; *Ikeda et al., 2001b*; *Holland et al., 2016*; *Merritt et al., 1999*). However, in vivo evidence for how each MAP2K contributes to DLK- and LZK-induced signal transduction cascade in neurons is limited (*Yang et al., 2015*; *Itoh et al., 2014*). Recent studies of T-cell activation have reported that the activity of MKK7 is regulated through alternative splicing of its exon 2, which encodes a small peptide within the JNK docking site in MKK7 (*Martinez et al., 2015*; *Figure 5—figure supplement 1A*). During T-cell activation, the RNA splicing factor CELF2 promotes skipping of

this exon, which favors the production of a short isoform of MKK7 that has high potency to activate JNK (*Martinez et al., 2015*; *Ajith et al., 2016*).

To test if this alternative splicing regulation of *Map2k7*, encoding mouse MKK7, has functional significance in neurons, we generated *Pvalb*<sup>Cre/+</sup>;*Celf2*<sup>fl/fl</sup>;Hipp11-DLK(iOE)/+ and *Pvalb*<sup>Cre/+</sup>;*Celf2*<sup>fl/fl</sup>; Hipp11-LZK(iOE)/+ mice, along with *Pvalb*<sup>Cre/+</sup>;*Celf2*<sup>fl/fl</sup> control mice (*Figure 5—figure supplement 1C*). By gross animal appearance and morphology of Purkinje cells, *Pvalb*<sup>Cre/+</sup>;*Celf2*<sup>fl/fl</sup> mice were indistinguishable from the control mice *Pvalb*<sup>Cre/+</sup> or *Celf2*<sup>fl/fl</sup> (*Figure 5—figure supplement 1D–H*). In *Pvalb*<sup>Cre/+</sup>;*Celf2*<sup>fl/fl</sup>;Hipp11-DLK(iOE)/+ mice Purkinje cell degeneration phenotypes were similar to those in *Pvalb*<sup>Cre/+</sup>;Hipp11-DLK(iOE)/+ mice (*Figure 5—figure supplement 1D–F*). Deletion of *Celf2* did not alter the expression levels of DLK (*Figure 5—figure supplement 1I*), nor the induction of p-c-Jun by increased DLK expression (*Figure 5—figure supplement 1J–L*). All pups of *Pvalb*<sup>Cre/+</sup>; *Celf2*<sup>fl/fl</sup>;Hipp11-DLK(iOE)/+ had smaller cerebellum (*Figure 5—figure supplement 1G*), showed morbidity and low body weight (*Figure 5—figure supplement 1H*), and died around P21. These data show that deletion of *Celf2* does not modify the phenotypes caused by increased DLK expression.

In contrast, *Pvalb*<sup>Cre/+</sup>;*Celf2*<sup>fl/fl</sup>;Hipp11-LZK(iOE)/+ mice showed significantly improved composite behavioral phenotypes, compared to *Pvalb*<sup>Cre/+</sup>;Hipp11-LZK(iOE)/+ from P30 to P120 (*Figure 5A–E*; *Video 2*), although the reduced body weight remained in *Pvalb*<sup>Cre/+</sup>;*Celf2*<sup>fl/fl</sup>;Hipp11-LZK(iOE)/+ mice (*Figure 5F*). At the cellular level, deletion of *Celf2* dramatically reduced Purkinje cell degeneration induced by elevated LZK expression, with significant improvement in the dendrite morphology at P120 (*Figure 5G–J*). Deletion of *Celf2* also significantly inhibited astrogliosis and microgliosis in P120 *Pvalb*<sup>Cre/+</sup>;*Celf2*<sup>fl/fl</sup>;Hipp11-LZK(iOE)/+ mice, compared to *Pvalb*<sup>Cre/+</sup>;Hipp11-LZK(iOE)/+ mice (*Figure 5—figure supplement 2A–D*). We detected a few microglia that appeared to contain dying Purkinje cells in *Pvalb*<sup>Cre/+</sup>;Hipp11-LZK(iOE)/+ mice but not in *Pvalb*<sup>Cre/+</sup>;*Celf2*<sup>fl/fl</sup>;Hipp11-LZK(iOE)/+ mice (*Figure 5—figure supplement 2C*; *Video 3*). Consistent with the suppression on Purkinje cell degeneration, deletion of *Celf2* also rescued the reduction of MAP2 and NF-200 in the molecular layer of cerebellum caused by elevated LZK expression (*Figure 5—figure supplement 3A–B*). Additionally, immunostaining to parvalbumin and neurofilament enabled the visualization of the basket cells, which form specialized structures, called pinceau, onto the axon initial segment (AIS) of Purkinje cells (*Palay and Chan-Palay, 1974*; *Ito, 1984*). In *Pvalb*<sup>Cre/+</sup>;Hipp11-LZK(iOE)/+ mice the pinceau were disorganized, which was suppressed by deletion of *Celf2* (*Figure 5—figure supplement 3B–C*).

To address whether CELF2 was involved in LZK signaling in Purkinje cells, we examined exon 2 splicing of *Map2k7*. By qRT-PCR analysis, we detected that *Celf2* deletion reduced mRNA ratio of the short isoform (*Map2k7-S*) to the long isoform (*Map2k7-L*) by ~20% in cerebellum of *Pvalb*<sup>Cre/+</sup>; *Celf2*<sup>fl/fl</sup>;Hipp11-LZK(iOE)/+ mice, compared to *Pvalb*<sup>Cre/+</sup>;Hipp11-LZK(iOE)/+ mice (*Figure 5—figure supplement 1B*). The overall expression levels of LZK in cerebellum were not altered by *Celf2* deletion (*Figure 6—figure supplement 1A–B*). We then immunostained for p-c-Jun in cerebellum of P120 mice. LZK overexpression induced p-c-Jun in most of Purkinje cells, and *Celf2* deletion significantly attenuated the intensity of p-c-Jun in Purkinje cells of *Pvalb*<sup>Cre/+</sup>;*Celf2*<sup>fl/fl</sup>;Hipp11-LZK(iOE)/+ mice (*Figure 6A–B*). The total number of dying cells, marked by TUNEL signals, in cerebellum of *Pvalb*<sup>Cre/+</sup>;*Celf2*<sup>fl/fl</sup>;Hipp11-LZK(iOE)/+ mice was significantly reduced, compared to that in *Pvalb*<sup>Cre/+</sup>;Hipp11-LZK(iOE)/+ mice (*Figure 6—figure supplement 1D–E*). A few TUNEL signals (approximately one cell/section) were co-localized with tdTomato-labeled Purkinje cells in *Pvalb*<sup>Cre/+</sup>;Hipp11-LZK(iOE)/+ mice, but not in *Pvalb*<sup>Cre/+</sup>;*Celf2*<sup>fl/fl</sup>;Hipp11-LZK(iOE)/+ mice (*Figure 6—figure supplement 1F*). Cleaved caspase-3 signals were rarely detected in Purkinje cells of *Pvalb*<sup>Cre/+</sup>;*Celf2*<sup>fl/fl</sup>; Hipp11-LZK(iOE)/+ mice, compared to *Pvalb*<sup>Cre/+</sup>;Hipp11-LZK(iOE)/+ mice (*Figure 6C–D*). We further assessed expression levels of Bcl-xL (B-cell lymphoma-extra Large), which as a full-length protein prevents caspase activation, but the cleaved protein product promotes apoptosis (*Gross et al., 1999*). Western blot analysis of cerebellar protein extracts from P21 mice when minimal Purkinje cell degeneration was detected in *Pvalb*<sup>Cre/+</sup>;Hipp11-LZK(iOE)/+ showed increased pro-apoptotic cleavage products of Bcl-xL, compared to control samples (*Figure 6—figure supplement 1A,C*). These data support a conclusion that *Celf2* deletion attenuates LZK-induced JNK signaling, and provide in vivo evidence that MKK7 is a functional mediator of LZK signaling in Purkinje cells.

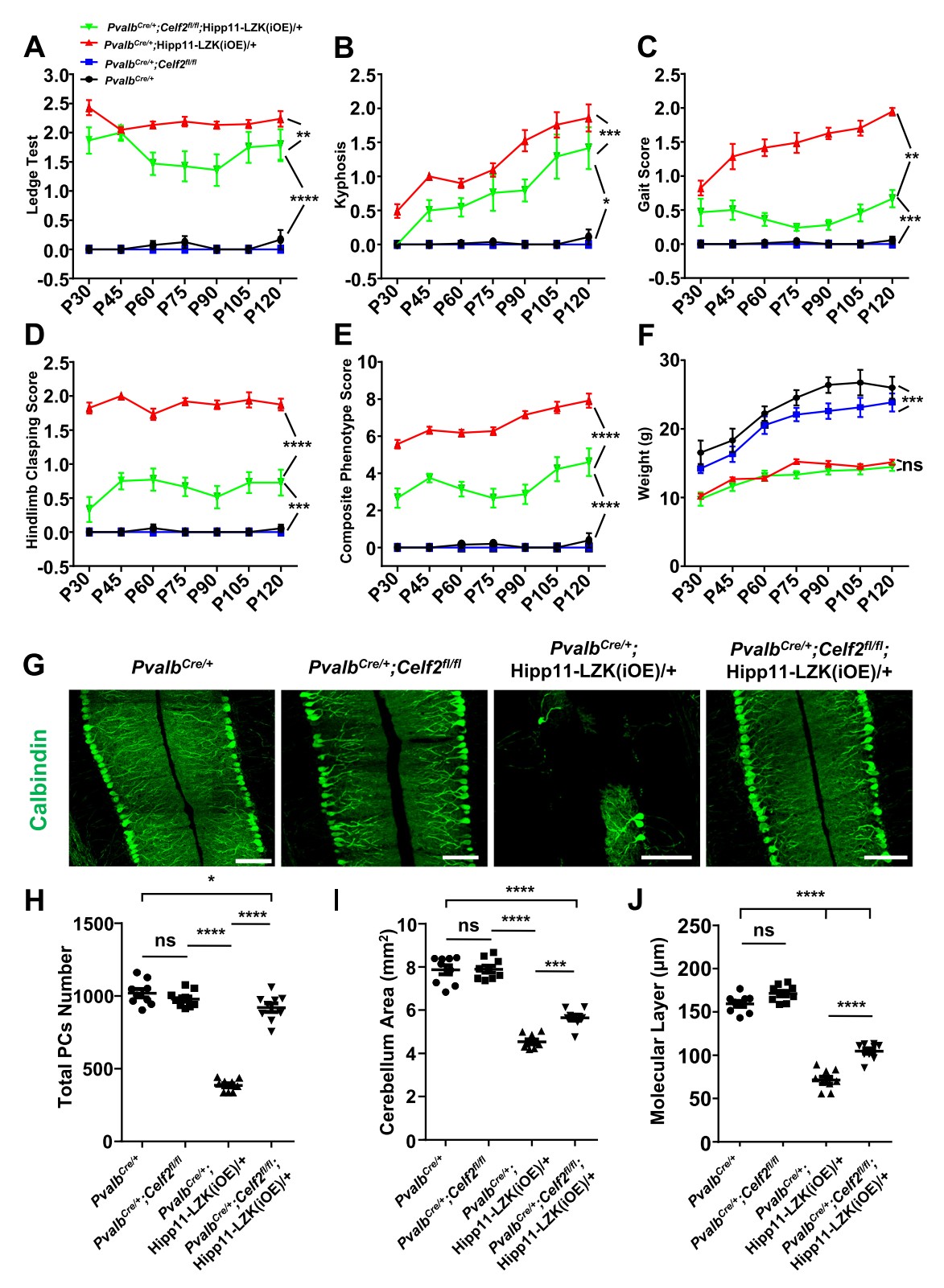

**Figure 5.** Deletion of the RNA splicing factor CELF2 using *Pvalb*^*Cre*^ rescues degeneration of Purkinje cells induced by elevated LZK expression. (A–E) Quantification of movement phenotypes of mice of indicated genotypes from P30 to P120. *Pvalb*^*Cre/+*^: n = 8 (P30), n = 17 (P45), n = 23 (P60), n = 19 (P75), n = 22 (P90), n = 12 (P105), n = 12 (P120); *Pvalb*^*Cre/+*^;*Celf2*^*fl/fl*^: n = 5 (P30), n = 8 (P45), n = 7 (P60), n = 6 (P75), n = 4 (P90), n = 4 (P105), n = 5 (P120); *Pvalb*^*Cre/+*^;Hipp11-LZK(iOE)/+: n = 21 (P30), n = 7 (P45), n = 20 (P60), n = 21 (P75), n = 26 (P90), n = 18 (P105), n = 19 (P120); *Pvalb*^*Cre/+*^;*Celf2*^*fl/fl*^;Hipp11-

*Figure 5 continued on next page*

*Figure 5 continued*

LZK(iOE)/+: n = 5 (P30), n = 8 (P45), n = 11 (P60), n = 11 (P75), n = 13 (P90), n = 8 (P105), n = 8 (P120). (F) Quantification of the body weight of mice of indicated genotypes from P30 to P120. $Pvalb^{Cre/+}$: n = 9 (P30), n = 9 (P45), n = 17 (P60), n = 14 (P75), n = 16 (P90), n = 10 (P105), n = 7 (P120); $Pvalb^{Cre/+}$; $Celf2^{fl/fl}$: n = 5 (P30), n = 5 (P45), n = 8 (P60), n = 6 (P75), n = 6 (P90), n = 5 (P105), n = 4 (P120); $Pvalb^{Cre/+}$;Hipp11-LZK(iOE)/+: n = 14 (P30), n = 16 (P45), n = 20 (P60), n = 14 (P75), n = 20 (P90), n = 17 (P105), n = 18 (P120); $Pvalb^{Cre/+}$;$Celf2^{fl/fl}$;Hipp11-LZK(iOE)/+: n = 7 (P30), n = 7 (P45), n = 12 (P60), n = 12 (P75), n = 14 (P90), n = 10 (P105), n = 10 (P120). Color representation for genotypes in B–F is the same as in A. (G) Representative images of Calbindin staining of cerebellar sections from P120 mice of genotypes indicated. Scale bars: 100 µm. (H) Quantification of total Purkinje cells in all cerebellar lobules at P120. (I) Quantification of cerebellum area at P120. (J) Quantification of the molecular layer thickness of P120 mice in cerebellar lobules V-VI. (H–J) n = 3 animals per genotype, three sections/animal. (A–F, H–J) Data shown are means ± SEM; Statistics: one-way ANOVA; ns, no significant; *, p<0.05; **, p<0.01; ***, p<0.001; ****, p<0.0001.

The online version of this article includes the following source data and figure supplement(s) for figure 5:

**Source data 1.** Score of ledge test of mice genotypes indicated from P30 to P120.
**Source data 2.** Score of kyphosis of mice genotypes indicated from P30 to P120.
**Source data 3.** Score of gait of mice genotypes indicated from P30 to P120.
**Source data 4.** Score of hindlimb clasping of mice genotypes indicated from P30 to P120.
**Source data 5.** Composite phenotype score of mice genotypes indicated from P30 to P120.
**Source data 6.** Mouse body weight of genotypes indicated from P30 to P120.
**Source data 7.** Total Purkinje cells in all cerebellar lobules of mice of genotypes indicated at P120.
**Source data 8.** Mouse cerebellum area of genotypes indicated at P120.
**Source data 9.** Molecular layer thickness in cerebellar lobules V-VI of P120 mice of genotypes indicated.
**Figure supplement 1.** CELF2 is a regulator of *Map2k7* in Purkinje cells.
**Figure supplement 2.** CELF2 deletion reduces astrogliosis and microgliosis associated with LZK in PV⁺ neurons.
**Figure supplement 3.** CELF2 deletion rescues reduced expression of MAP2 and NF-200 and pinceau disorganization induced by elevated LZK expression.

## DLK and LZK can induce Purkinje cell degeneration independent of each other

DLK and LZK have a nearly identical kinase domain, and are reported to bind and be co-immunoprecipitated from mouse brain (*Pozniak et al., 2013*). Recent studies have also shown that in injured RGCs or DRGs the two kinases have redundant or synergistic interactions (*Welsbie et al., 2017*; *Summers et al., 2020*). We next addressed whether the Purkinje cell degeneration caused by elevating DLK or LZK activity depends on the presence of one another by analyzing $Pvalb^{Cre/+}$; $Map3k13^{KO/KO}$;Hipp11-DLK(iOE)/+ and $Pvalb^{Cre/+}$;$Map3k12^{fl/fl}$;Hipp11-LZK(iOE)/+ mice.

In $Pvalb^{Cre/+}$;$Map3k13^{KO/KO}$;Hipp11-DLK(iOE)/+ mice, Purkinje cell degeneration and astrogliosis proceeded temporally and spatially similar to that of $Pvalb^{Cre/+}$;Hipp11-DLK(iOE)/+ (*Figure 7A–E*). The elevated p-c-Jun levels in Purkinje cells of $Pvalb^{Cre/+}$;Hipp11-DLK(iOE)/+ mice at P10 were not affected by deleting *Mapk3k13*, encoding LZK (*Figure 7—figure supplement 1A–B*). All pups of $Pvalb^{Cre/+}$;$Map3k13^{KO/KO}$;Hipp11-DLK(iOE)/+ died by P21, with similar morbidity as $Pvalb^{Cre/+}$; Hipp11-DLK(iOE)/+. Conversely, the $Pvalb^{Cre/+}$;$Map3k12^{fl/fl}$;Hipp11-LZK(iOE)/+ mice resembled $Pvalb^{Cre/+}$;Hipp11-LZK(iOE)/+ mice in the slow degeneration of Purkinje cells. Both $Pvalb^{Cre/+}$; $Map3k12^{fl/fl}$;Hipp11-LZK(iOE)/+ and $Pvalb^{Cre/+}$;Hipp11-LZK(iOE)/+ mice had low body weight (*Figure 7—figure supplement 1C*) and small cerebellum area at P60 (*Figure 7I*), compared to the control mice. By immunostaining with Calbindin and GFAP antibodies, we observed that removing *Map3k12,* encoding DLK, did not alter Purkinje cell degeneration or astrogliosis caused by LZK expression (*Figure 7F,H,J*). The levels of p-c-Jun induced by LZK expression in Purkinje cells remained comparable, with or without endogenous DLK (*Figure 7—figure supplement 1D–E*). These data show that targeted activation of each kinase induces Purkinje cell degeneration largely independent of each other.

## Discussion

In this study, we have used cerebellar Purkinje cells to gain a systematic understanding of the function of two closely related kinases, DLK (MAP3K12) and LZK (MAP3K13), that have emerged as key players in neural protection under injury and disease (*Asghari Adib et al., 2018*; *Jin and Zheng, 2019*; *Farley and Watkins, 2018*). We employed both conditional KO and transgenic Cre-inducible expression mice to manipulate levels of DLK and LZK expression. We find that while deleting one or

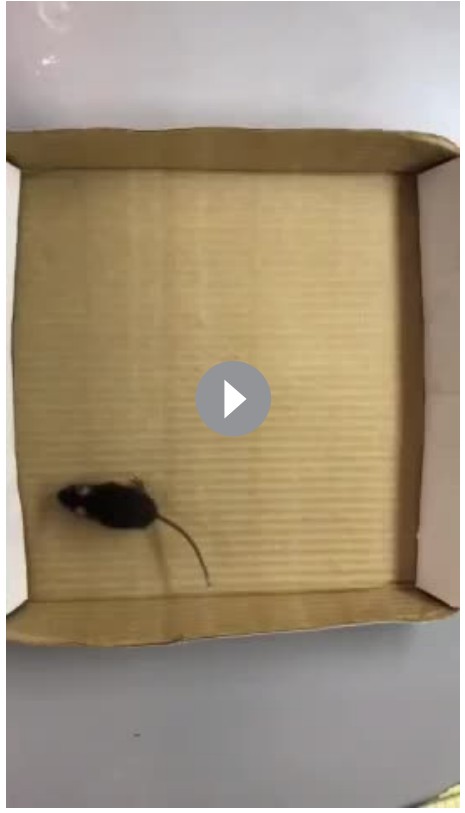

**Video 2.** Locomotor deficits in P120 *Pvalb*^Cre/+;Hipp11-LZK(iOE)/+ mice. A representative P120 old *Pvalb*^Cre/+;Hipp11-LZK(iOE)/+ mouse exhibited severe limp while walking and tremors.
https://elifesciences.org/articles/63509#video2

stitutive knockout mice the developing spinal cord motor neurons and periphery sensory neurons show increased survival (*Itoh et al., 2011*; *Sengupta Ghosh et al., 2011*). These studies support the general notion that these two kinases exert cell-type, stage- and context-dependent effects. Additionally, we observed body weight loss, motor deficits and early mortality associated with *Pvalb*^Cre/+;Hipp11-DLK(iOE)/+ pups. Parvalbumin-Cre is expressed widely throughout the nervous system, including many interneurons in the brain and sensory neurons in periphery nervous system (*Hippenmeyer et al., 2005*). Purkinje cells are not essential for animal viability. While our data do not address the cellular and/or circuit basis of these gross phenotypes, we infer that the grossly abnormality and lethality of *Pvalb*^Cre/+;Hipp11-DLK(iOE)/+ pups are unlikely the consequence of Purkinje cell death alone. The underlying basis would be of interest in future studies.

DLK and LZK share a kinase domain that is ~90% identical and can activate the JNK

both kinases in Purkinje cells postnatally did not affect their development, elevating expression of DLK or LZK, which leads to their activation, causes Purkinje cell degeneration. Our Cre-inducible DLK and LZK transgenes have the same design and are inserted in the same genomic locus to avoid position effect on transgene expression. Despite the similarly targeted transgenes, we found that DLK elevation triggers rapid degeneration of Purkinje cells, while LZK elevation causes slow degeneration (*Figure 7—figure supplement 2*). Each kinase activates JNK signaling, measured by increased phosphorylated c-Jun, and induces apoptosis. Each kinase can induce neuron degeneration in the absence of the other. Importantly, we show that deletion of mRNA splicing factor CELF2 strongly attenuates Purkinje cell degeneration caused by LZK, but not DLK, providing further evidence for a signaling pathway-specific effect for each kinase rather than a generic, secondary effect of overexpressing any kinase. These data support the utility of our transgenic mice for dissecting cell-type specific roles of these kinases and their signaling pathways.

Our findings that DLK and LZK are dispensable for postnatal development of Purkinje cells are in line with the report that adult knockout of DLK exhibit nearly normal function (*Le Pichon et al., 2017*; *Pozniak et al., 2013*), but are in contrast to the previous studies that in DLK con-

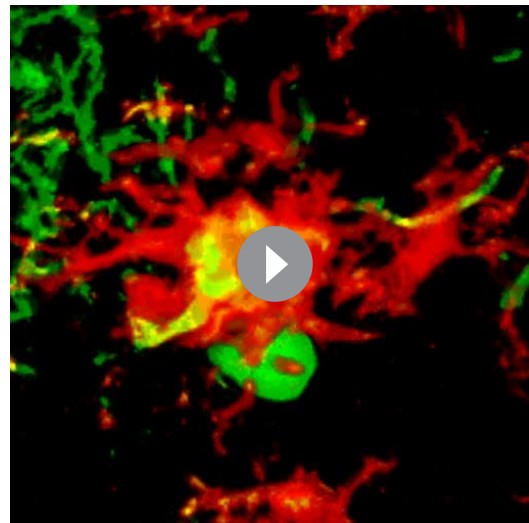

**Video 3.** Calbindin^+ staining materials are present inside of IBA1^+ cells in P120 *Pvalb*^Cre/+;Hipp11-LZK(iOE)/+ mice. A video showing the 3D reconstruction image of Calbindin^+ staining materials in the microglia or macrophage cell body.
https://elifesciences.org/articles/63509#video3

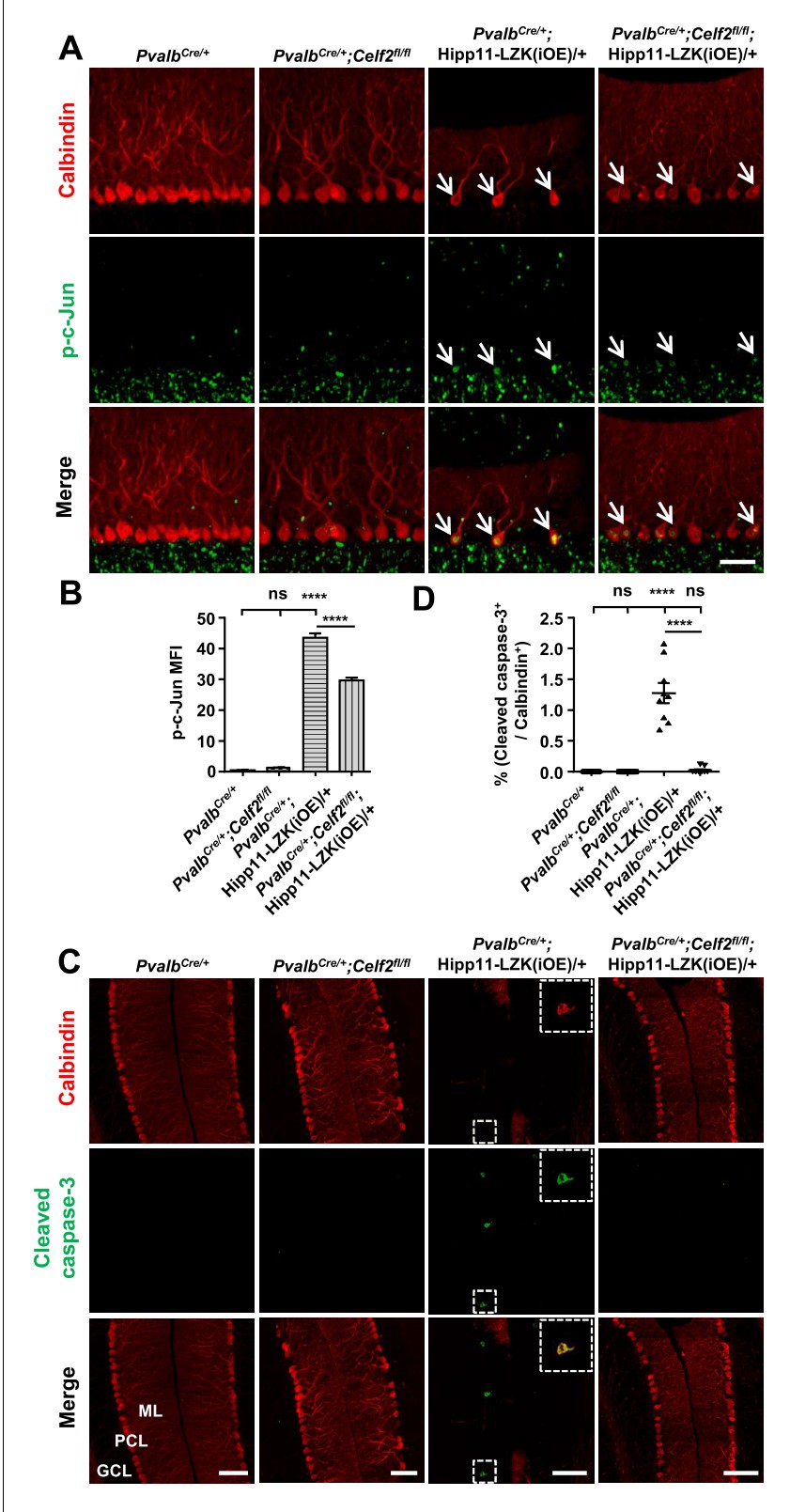

**Figure 6.** Deletion of CELF2 in *Pvalb*^Cre/+^;Hipp11-LZK(iOE)/+ mice reduces levels of p-c-Jun and apoptosis in Purkinje cells. (**A**) Representative images of Purkinje cells, co-immunostained for p-c-Jun and Calbindin, in P120 mice of genotypes indicated. Arrows point to p-c-Jun immunostaining signals in Purkinje cells. ML: Molecular Layer; PCL: Purkinje Cell Layer; GCL: Granule Cell Layer. Scale bar: 100 μm. (**B**) Quantification of the p-c-Jun levels

*Figure 6 continued on next page*

*Figure 6 continued*

in Purkinje cells of P120 mice. MFI: mean of fluorescence intensity. (**C**) Representative images of Purkinje cells, co-immunostained for cleaved caspase-3 and Calbindin, in P120 mice of genotypes indicated, with enlarged boxes showing cleaved caspase-3$^+$ signal in Calbindin-labeled Purkinje cells. ML: Molecular Layer; PCL: Purkinje Cell Layer; GCL: Granule Cell Layer. Scale bars: 100 μm. (**D**) Quantification of the percentage of cleaved caspase-3$^+$ Purkinje cells in total number of Purkinje cells of P120 mice of genotypes indicated. (**B, D**) n = 3 animals per genotype, three sections/animal; data shown are means ± SEM. Statistics: one-way ANOVA; ns, no significant; ****, p<0.0001.

The online version of this article includes the following source data and figure supplement(s) for figure 6:

**Source data 1.** p-c-Jun levels in Purkinje cells of P120 mice of genotypes indicated.
**Source data 2.** Percentage of cleaved caspase-3$^+$ Purkinje cells in total number of Purkinje cells of P120 mice of genotypes indicated.
**Figure supplement 1.** Additional evidence of apoptosis in Purkinje cell degeneration caused by elevated LZK expression.

signaling pathway through two MAPKK, MKK4, or MKK7 (*Hirai et al., 2011*; *Le Pichon et al., 2017*; *Huang et al., 2017*; *Ikeda et al., 2001a*; *Chen et al., 2016b*; *Ikeda et al., 2001b*; *Holland et al., 2016*; *Merritt et al., 1999*). While some in vivo studies have supported that MKK4 is a major mediator for DLK (*Yang et al., 2015*; *Itoh et al., 2014*), currently little is known about which MAPKK mediates LZK signaling. Our data show that elevating DLK expression in Purkinje cells leads to robust JNK signaling, compared to LZK. The observation that deletion of CELF2 did not affect any phenotypes caused by elevated DLK expression could be due to a combination of the strong JNK activation and the rapid time course of cell death. In contrast, deletion of CELF2 significantly reduced the activation of c-Jun and almost completely rescued the Purkinje cell degeneration caused by elevated LZK expression. These data are consistent with the role of CELF2 in regulating alternative splicing of *Map2k7* (encoding MKK7) (*Martinez et al., 2015*), and support that MKK7 is a functional downstream kinase for LZK in vivo. The observation that CELF2 deletion led to a more prominent reduction of cleaved caspase-3 than p-c-Jun levels implies that apoptotic pathway downstream of LZK is more sensitive to MKK7 activity, while other unidentified MAP2K may also act downstream of LZK.

Numerous studies have revealed roles of DLK in axon growth, regeneration, and/or degeneration (*Tedeschi and Bradke, 2013*; *Jin and Zheng, 2019*; *Asghari Adib et al., 2018*). DLK is also known to regulate microtubule stability (*Simard-Bisson et al., 2017*; *Valakh et al., 2015*; *Hirai et al., 2011*), and several microtubule-associated proteins such as SCG10, DCX, and MAP2 are JNK substrates (*Chang et al., 2003*; *Gdalyahu et al., 2004*; *Tararuk et al., 2006*; *Björkblom et al., 2005*). Our analyses on Purkinje cells are consistent with the known roles of DLK and JNK regulation of microtubule associated proteins. Comparatively, not much is known about the roles of LZK in neurons. We show that deleting LZK does not disrupt Purkinje cell morphology, but increasing LZK expression caused reduced neurofilament levels in the molecular layer of cerebellum and disorganization of the pinceau at the axon initial segment of Purkinje cell. LZK expression also significantly decreased MAP2 levels in dendrites of Purkinje cells, and CELF2 deletion restored MAP2 levels only in *Pvalb^{Cre/+}*;*Celf2^{fl/fl}*;Hipp11-LZK(iOE)/+ mice. These data begin to uncover the cellular effects of LZK signaling in neurons.

There is mounting evidence in the literature on the upregulation of the DLK and LZK signaling pathway in CNS injury and neurodegeneration (*Watkins et al., 2013*; *Welsbie et al., 2013*; *Joy et al., 2019*; *Le Pichon et al., 2017*; *Huang et al., 2017*; *Chen et al., 2018*), indicating that altered signaling of this pathway may be a prevalent feature in CNS injury and diseases. Along this line, genetic studies of DLK in both invertebrate and vertebrate species have revealed prominent developmental defects with DLK activation rather than inactivation (*Lewcock et al., 2007*; *Nakata et al., 2005*; *Collins et al., 2006*; *Grill et al., 2016*). Our findings show that DLK-induced signal transduction cascade triggers a strong or rapid cellular response, which may be more suited under conditions of traumatic injury or insults, while LZK induces modest activation of JNK and apoptosis, which may manifest in chronic neurodegenerative diseases. Besides the kinase domain and leucine-zipper domain, both MAP3Ks have large uncharacterized C-terminus, which may play significant roles in regulating the signaling strength of each protein. As DLK and LZK activity exhibits high cell-type and context-dependent specificity, our transgenic mice offer valuable gain of function

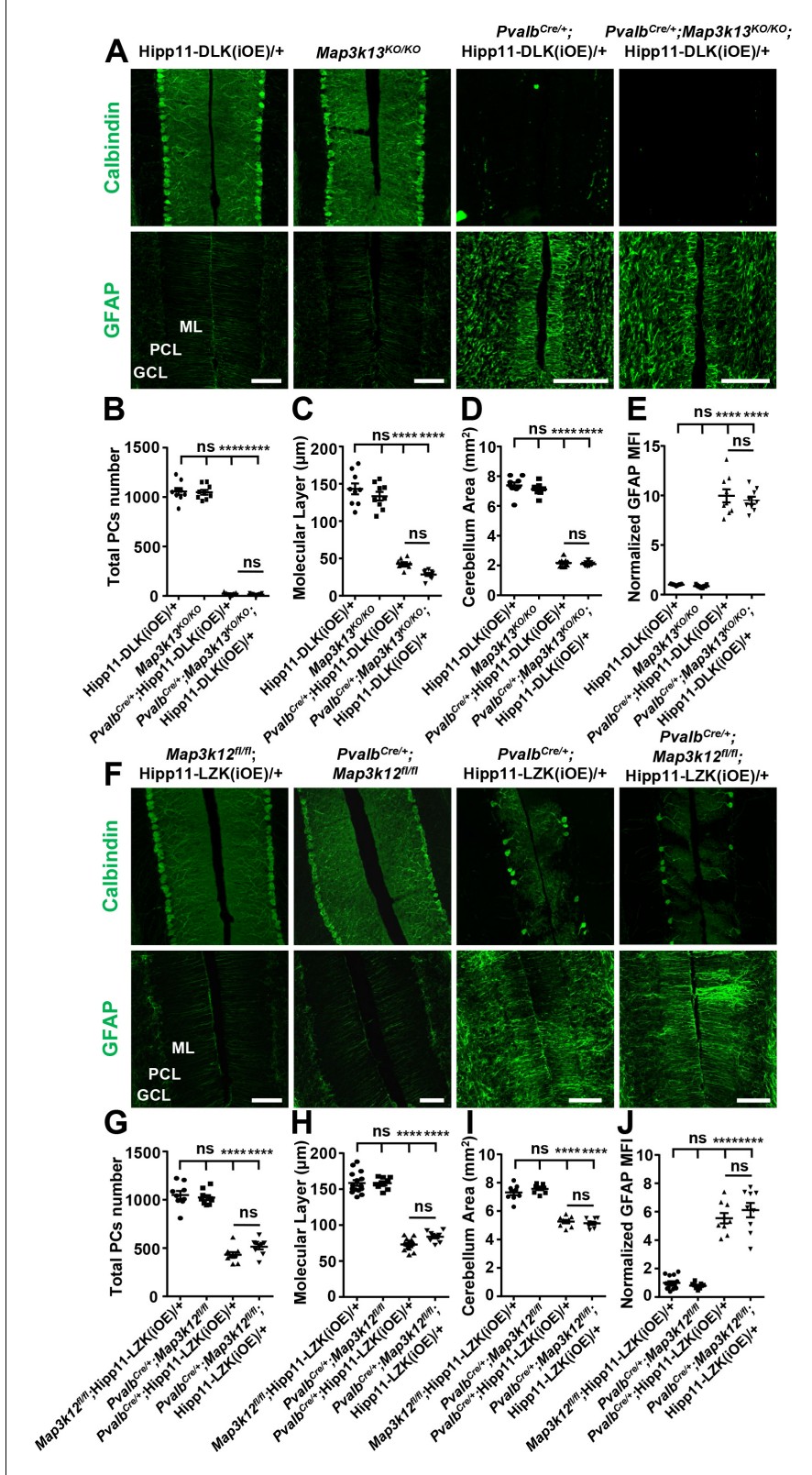

**Figure 7.** DLK and LZK can induce Purkinje cell degeneration independent of each other. (**A**) Representative images of immunostainning for Calbindin (upper panels) and GFAP (lower panels) of cerebellar sections of P21 mice of genotypes indicated. ML: Molecular Layer; PCL: Purkinje Cell Layer; GCL: Granule Cell Layer. Scale bars: 100 μm. (**B**) Quantification of total number of Purkinje cells in all cerebellar lobules at P21. (**C**) Quantification of the

*Figure 7 continued on next page*

*Figure 7 continued*

molecular layer thickness in cerebellar lobules V-VI of P21 mice. (**D**) Quantification of the cerebellum area of P21 mice. (**E**) GFAP staining intensity is quantified and normalized to Hipp11-DLK (iOE)/+ in cerebellum of P21 mice. MFI: mean of fluorescence intensity. (**F**) Representative images of immunostaining for Calbindin (upper panels) and GFAP (lower panels) of cerebellar sections of P60 mice of genotypes indicated. ML: Molecular Layer; PCL: Purkinje Cell Layer; GCL: Granule Cell Layer. Scale bars: 100 μm. (**G**) Quantification of total number of Purkinje cells in all cerebellar lobules of P60 mice. (**H**) Quantification of the molecular layer thickness in cerebellar lobules V-VI of P60 mice. *Map3k12$^{fl/fl}$*;Hipp11-LZK(iOE)/+ and *Pvalb$^{Cre/+}$*;Hipp11-LZK(iOE)/+: n = 5 animals, three sections/animal; *Pvalb$^{Cre/+}$*;*Map3k12$^{fl/fl}$* and *Pvalb$^{Cre/+}$*;*Map3k12$^{fl/fl}$*;Hipp11-LZK(iOE)/+: n = 3 animals, three sections/animal. (**I**) Quantification of the cerebellum area of P60 mice. (**J**) GFAP staining intensity is quantified and normalized to *Map3k12$^{fl/fl}$*;Hipp11-LZK(iOE)/+ in cerebellum of P60 mice. MFI: mean of fluorescence intensity. *Map3k12$^{fl/fl}$*;Hipp11-LZK(iOE)/+: n = 5 animals, three sections/animal; *Pvalb$^{Cre/+}$*;*Map3k12$^{fl/fl}$*, *Pvalb$^{Cre/+}$*;Hipp11-LZK(iOE)/+ and *Pvalb$^{Cre/+}$*;*Map3k12$^{fl/fl}$*;Hipp11-LZK(iOE)/+: n = 3 animals, three sections/animal. (**B–E, G, I**) n = 3 animals per genotype, three sections/animal. (**B–E, G–J**) Data shown are means ± SEM; Statistics: one-way ANOVA; ns, no significant; ****, p<0.0001.

The online version of this article includes the following source data and figure supplement(s) for figure 7:

**Source data 1.** Total number of Purkinje cells in all cerebellar lobules of P21 mice of genotypes indicated.
**Source data 2.** Molecular layer thickness in cerebellar lobules V-VI of P21 mice of genotypes indicated.
**Source data 3.** Cerebellum area of P21 mice of genotypes indicated.
**Source data 4.** Normalized GFAP levels in cerebellum of P21 mice of genotypes indicated.
**Source data 5.** Total number of Purkinje cells in all cerebellar lobules of P60 mice of genotypes indicated.
**Source data 6.** Molecular layer thickness in cerebellar lobules V-VI of P60 mice of genotypes indicated.
**Source data 7.** Cerebellum area of P60 mice of genotypes indicated.
**Source data 8.** Normalized GFAP levels in cerebellum of P60 mice of genotypes indicated.
**Figure supplement 1.** Levels of p-c-Jun induced by elevated expression of DLK or LZK is not affected by loss of LZK or DLK, respectively.
**Figure supplement 2.** Summary of Purkinje cell death observed in the mice with elevated expression of two MAP3Ks.

models to study their signaling pathways with the ease for temporal and spatial manipulation. The knowledge learned will advance our understanding of how diverse neuronal types respond to insults to the nervous system.

# Materials and methods

## Key resources table

| Reagent type (species) or resource | Designation | Source or reference | Identifiers | Additional information |
|---|---|---|---|---|
| Genetic reagent (*M. musculus*) | B6.129P2-*Pvalb$^{tm1(cre)Arbr}$*/J (*Pvalb$^{Cre}$*) | Jackson Laboratory | Stock #: 017320 RRID:MGI_3590684 | |
| Genetic reagent (*M. musculus*) | C57BL/6J | Jackson Laboratory | Stock #: 000664 RRID:MGI_3028467 | |
| Genetic reagent (*M. musculus*) | Inducible DLK overexpression: Hipp11-DLK(iOE) | This paper | | housed in UCSD vivarium |
| Genetic reagent (*M. musculus*) | Inducible LZK overexpression: Hipp11-LZK(iOE) | This paper | | housed in UCSD vivarium |
| Genetic reagent (*M. musculus*) | Conditional DLK knockout (*Map3k12$^{fl}$*) | *Chen et al., 2016b* | | housed in UCSD vivarium |
| Genetic reagent (*M. musculus*) | Conditional LZK knockout (*Map3k13$^{fl}$*) | *Chen et al., 2016b* | | housed in UCSD vivarium |
| Genetic reagent (*M. musculus*) | Conditional *Celf2* knockout (*Celf2$^{fl}$*) | *Chen et al., 2016a* | | housed in UCSD vivarium |
| Genetic reagent (*M. musculus*) | CRISPR LZK knockout (*Map3k13$^{KO}$*) | This paper | | housed in UCSD vivarium |

*Continued on next page*

*Continued*

| Reagent type (species) or resource | Designation | Source or reference | Identifiers | Additional information |
|---|---|---|---|---|
| Antibody | Anti-MAP3K12 (rabbit polyclonal) | Sigma-Aldrich | Cat. #: SAB2700169 RRID:AB_2714162 | WB (1:2000), IF (1:250) |
| Antibody | Anti-MAP3K13 (rabbit polyclonal) | Sigma-Aldrich | Cat. #: HPA016497 RRID:AB_10670027 | WB (1:300), IF (1:200), IP (2 μg per sample) |
| Antibody | Anti-Actin (Clone C4) (mouse monoclonal) | MP Biomedicals | Cat. #: 08691001 RRID:AB_2335127 | WB (1:10,000) |
| Antibody | Anti-Bcl-xL (54H6) (rabbit monoclonal) | Cell signaling | Cat. #: 2764 RRID:AB_2228008 | WB (1:2000) |
| Antibody | Anti-FLAG (rabbit polyclonal) | Millipore | Cat. #: F7425 RRID:AB_439687 | WB (1:500) |
| Antibody | Sheep anti-mouse IgG (sheep polyclonal, HRP conjugated) | GE healthcare | Cat. #: NA931 RRID:AB_772210 | WB (1:5000) |
| Antibody | Donkey anti-rabbit IgG (donkey polyclonal, HRP conjugated) | GE healthcare | Cat. #: NA934 RRID:AB_772206 | WB (1:5000) |
| Antibody | Anti-Iba1 (rabbit polyclonal) | Wako | Cat. #: 019–19741 RRID:AB_839504 | IF (1:1000) |
| Antibody | Anti-Neurofilament 200 (Phos. and Non-Phos.) (mouse monoclonal) | Sigma-Aldrich | Cat. #: N0142 RRID:AB_477257 | IF (1:200) |
| Antibody | Anti-Phospho-c-Jun (Ser73) (D47G9) XP (rabbit monoclonal) | Cell signaling | Cat. #: 3270 RRID:AB_2129575 | IF (1:200) |
| Antibody | Anti-Cleaved Caspase-3 (Asp175) (rabbit polyclonal) | Cell signaling | Cat. #: 9661 RRID:AB_2341188 | IF (1:200) |
| Antibody | Anti-Calbindin (D1I4Q) XP (rabbit monoclonal) | Cell signaling | Cat. #: 13176 RRID:AB_2687400 | IF (1:500) |
| Antibody | Anti-Calbindin D-28k (mouse monoclonal) | Swant | Cat. #: 300 RRID:AB_10000347 | IF (1:500) |
| Antibody | Anti-GFAP (2.2B10) (rat monoclonal) | Thermo Fisher Scientific | Cat. #: 13–0300 RRID:AB_2532994 | IF (1:500) |
| Antibody | Anti-MAP2 (chicken polyclonal) | Abcam | Cat. #: ab5392 RRID:AB_2138153 | IF (1:500) |
| Antibody | Anti-Parvalbumin (guinea pig polyclonal) | Synaptic systems | Cat. #: 195004 RRID:AB_2156476 | IF (1:500) |
| Antibody | Anti-tdTomato (goat polyclonal) | SICGEN | Cat. #: AB8181-200 RRID:AB_2722750 | IF (1:500) |
| Antibody | Goat anti-mouse IgG (H+L) (goat polyclonal, Alexa Fluor 647 conjugated) | Invitrogen | Cat. #: A21236 RRID:AB_2535805 | IF (1:500) |
| Antibody | Goat anti-guinea Pig IgG (H+L) (goat polyclonal, Alexa Fluor 488 conjugated) | Invitrogen | Cat. #: A11073 RRID:AB_2534117 | IF (1:500) |
| Antibody | Goat anti-rabbit IgG (H+L) (goat polyclonal, Alexa Fluor 488 conjugated) | Invitrogen | Cat. #: A11008 RRID:AB_143165 | IF (1:500) |
| Antibody | Goat anti-rabbit IgG (H+L) (goat polyclonal, Alexa Fluor 647 conjugated) | Invitrogen | Cat. #: A21245 RRID:AB_2535813 | IF (1:500) |
| Antibody | Goat anti-mouse IgG (H+L) (goat polyclonal, Alexa Fluor 488 conjugated) | Invitrogen | Cat. #: A11001 RRID:AB_2534069 | IF (1:500) |

*Continued on next page*

*Continued*

| Reagent type (species) or resource | Designation | Source or reference | Identifiers | Additional information |
|---|---|---|---|---|
| Antibody | Goat anti-rat IgG (H+L) (goat polyclonal, Alexa Fluor 488 conjugated) | Invitrogen | Cat. #: A11006 RRID:AB_141373 | IF (1:500) |
| Antibody | Goat anti-chicken IgG (H+L) (goat polyclonal, Alexa Fluor 647 conjugated) | Invitrogen | Cat. #: A21449 RRID:AB_1500594 | IF (1:500) |
| Recombinant DNA reagent | pBT378-LSL-3X Flag-DLK-T2A-td Tomato (plasmid) | This paper | | Construct Hipp11-DLK(iOE) mice |
| Recombinant DNA reagent | pBT378-LSL-1X Flag-LZK-T2A-td Tomato (plasmid) | This paper | | Construct Hipp11-LZK(iOE) mice |
| Sequence -based reagent | sgRNA2-Fw | This paper | | CACCGTGGCACTACAGGTCACATAC |
| Sequence -based reagent | sgRNA2-Re | This paper | | AAACGTATGTGACCTGTAGTGCCAC |
| Sequence -based reagent | sgRNA5-Fw | This paper | | CACCGGACCTCGTACAGCTGTCCGT |
| Sequence -based reagent | sgRNA5-Re | This paper | | AAACACGGACAGCTGTACGAGGTCC |
| Sequence -based reagent | sgRNA12-Fw | This paper | | CACCGACTCCAGTATAGCCTCGATG |
| Sequence -based reagent | sgRNA12-Re | This paper | | AAACCATCGAGGCTATACTGGAGTC |
| Commercial assay or kit | SuperScript IV First-Strand Synthesis System | Invitrogen | 18091050 | |
| Commercial assay or kit | iQ SYBR Supermix | Bio-Rad | 170–8882 | |
| Commercial assay or kit | Surveyor Mutation Detection Kit | IDT | 706020 | |
| Commercial assay or kit | MEGAscript T7 Transcription Kit | Invitrogen | AMB13345 | |
| Commercial assay or kit | MEGAclear-96 Transcription Clean-Up Kit | Invitrogen | AM1909 | |
| Commercial assay or kit | H and E staining Kit | Abcam | ab245880 | |
| Commercial assay or kit | Pierce BCA protein assay kit | Thermo Fisher Scientific | 23225 | |
| Commercial assay or kit | DeadEnd Fluorometric TUNEL System | Promega | G3250 | |
| Software, algorithm | ImageJ software | ImageJ (https://imagej.net/) | RRID:SCR_003070 | |
| Software, algorithm | GraphPad Prism software | GraphPad Prism (http://www.graphpad.com/) | RRID:SCR_002798 | Version 6.0 |
| Software, algorithm | CFX Manager | CFX Manager (http://www.bio-rad.com/en-eh/product/cfx-manager-software) | RRID:SCR_017251 | |
| Software, algorithm | NDP.view2 Viewing software | Hamamatsu Photonics (https://www.hamamatsu.com/us/en/product/type/U12388-01/index.html) | RRID:SCR_017105 | |
| Other | DAPI stain | Invitrogen | D1306 | (14.3 µM) |

## Mice

All animal protocols were approved by the Animal Care and Use Committee of the University of California San Diego. Wild-type C57BL/6J mice (Stock No: 000664; RRID: MGI_3028467) and *Pvalb*<sup>*Cre*</sup> mice (Stock No: 017320; RRID: MGI_3590684) were purchased from The Jackson Laboratory.

*Map3k13* knockout mice were generated in the UCSD Transgenic and Knockout Mouse Core, using CRISPR-Cas9 technology (*Ran et al., 2013*). Briefly, sgRNA sequences targeted to the kinase domain were designed using online tools (http://crispr.mit.edu) (Key Resources Table). The selected sgRNAs were annealed, and then cloned into PX330 backbone digested with BbsI. Effectiveness of sgRNAs was tested using Surveyor nuclease assay (Surveyor Mutation Detection Kit, IDT, 706020). To make sgRNAs, DNA fragments containing T7 promoter followed by sgRNA were first amplified using primers YJ12532-12535. The purified DNAs were then in vitro transcribed using MEGAscript T7 Transcription Kit (Invitrogen, AMB13345), and the resulting transcripts were purified using MEGA-clear-96 Transcription Clean-Up Kit (Invitrogen, AM1909). The sgRNAs and Cas9 mRNA were injected into zygotes from C57BL6, which were then implanted into the CD1 surrogate mothers. Two KO mouse lines were obtained and the one containing a deletion of the entire kinase domain was used in this study. *Map3k13*<sup>*fl*</sup> and *Map3k12*<sup>*fl*</sup> mice were reported in *Chen et al., 2016b*. *Map3k12*<sup>*fl*</sup> mice were a kind gift of Dr. Lawrence B. Holzman (Univ. Penn). *Celf2*<sup>*fl*</sup> mice were described previously (*Chen et al., 2016a*). Primers for genotypes are listed in *Table 1*.

Transgenic mice for inducible expression, Hipp11-DLK(iOE) and Hipp11-LZK(iOE), were made by Applied StemCell, Inc (Milpitas, CA), using TARGATT Technology (*Tasic et al., 2011*). A mixture of plasmid pBT378-LSL-3X Flag-DLK-T2A-tdTomato, or pBT378-LSL-1X Flag-LZK-T2A-tdTomato DNA, and in vitro transcribed ϕC31 integrase mRNA was microinjected into the pronucleus of zygotes from a FVB strain that has the Att recombination landing site inserted in *Hipp11* locus (*Hippenmeyer et al., 2010*), which were then implanted into the CD1 surrogate mothers. The founder heterozygous mice were bred three times to pure C57BL/6J (Stock No: 000664; RRID: MGI_3028467) background.

## Histology and immunocytochemistry

Mice were transcardially perfused with 0.9% saline solution and then 4% paraformaldehyde (PFA) in phosphate-buffered saline (PBS) (pH 7.2–7.4). Brains were dissected, post-fixed in 4% PFA overnight at 4°C, and then transferred to 30% sucrose in PBS, prior to embedding using O.C.T compound (Fisher Healthcare, 4585) on dry ice. A total of 25-μm-thick sagittal sections were collected on a cryostat (Leica, CM1850) into PBS with 0.01% NaN$_3$. For histology analysis, free floating tissue sections were loaded to the slide, and then sequentially stained with hematoxylin and eosin (H and E staining Kit, abcam, ab245880). For immunostaining, free floating tissue sections were washed twice in PBS with 0.2% Triton X-100, blocked with 5% goat serum in PBS with 0.4% Triton X-100 for 2 hr at room temperature, then incubated with primary antibodies (Key Resources Table) diluted in PBS with 0.2% Triton X-100 and 2% goat serum overnight at 4°C. Alexa Fluor 488-conjugated and Alexa Fluor 647-conjugated secondary antibodies (Key Resources Table) were incubated for 2 hr at room temperature, followed by staining with DAPI (14.3 μM in PBS, Thermo Fisher Scientific, D1306) for 10 min. Stained sections were mounted with prolong diamond antifade mountant (Thermo Fisher Scientific, P36970).

## TUNEL staining

The DeadEnd Fluorometric TUNEL System (Promega, G3250) was used with a modified protocol. Free floating tissue sections were washed twice in PBS, and then loaded to the slide. The slides were dried at 65°C for 5 min, then immersed into PBS with 0.5% Triton X-100 and incubated at 85°C for 20 min, followed by three times rinsing with PBS. The slides were incubated with equilibration buffer at room temperature for 5 min, and then incubated with reaction mix (equilibration buffer: nucleotide mix: rTdT enzyme = 45:5:1) at 37°C for 1 hr in a humidity chamber. The reactions were stopped by incubating the slide with 2X saline-sodium citrate (SSC) buffer at room temperature for 10 min. After three washes with PBS, the slides were incubated with DAPI (14.3 μM in PBS) at room temperature for 15 min, followed by three washes with ddH$_2$O, and then mounted with prolong diamond antifade mountant (Thermo Fisher Scientific, P36970).

**Table 1.** Primers used in this study.

| Designation | 5'−3' | PCR Products (bp) | Purpose |
|---|---|---|---|
| YJ12516 | CGACCTCAACTTGGATATCAGCC | 148 (*Map2k7-L*) | RT-PCR |
| YJ12517 | GGAGCTCTCTGAGGATGGTGAGC | 100 (*Map2k7-S*) | |
| YJ12518 | GCTTCTTTGCAGCTCCTTCGT | 201 (*Actin*) | RT-PCR |
| YJ12519 | CCTTCTGACCCATTCCCACC | | |
| YJ12520 | TGGAGGAGGACAAACTGGTCA | 323 (Wild-type) | Genotyping |
| YJ12521 | TTCCCTTTCTGCTTCATCTTGC | ~500 (Hipp11-DLK(iOE)) | |
| YJ12522 | CATATATGGGCTATGAACTAATGACCCCGT | ~600 (Hipp11-LZK(iOE)) | |
| YJ12523 | GATGATTGCTAGTCATGGAGTAGTAGG | 350 (Wild-type) | Genotyping |
| YJ12524 | GGTGGTGTTATCATAGTTCCATCATG | 500 (*Map3k12^fl/fl*) | |
| YJ12525 | ACATCGGATCTGAAGACAGCCAGG | ~500 (Wild-type) | Genotyping |
| YJ12526 | AGGTGCGTTTTCATTCTTCTGGACC | ~600 (*Map3k13^fl/fl*) | |
| YJ12527 | GAGGTGTCTGCCGAACT | 370 (Wild-type) | Genotyping |
| YJ12528 | CACTCAGTCCCTGTTTGTAA | 470 (*Celf2^fl/fl*) | |
| YJ12529 | TGGCAGTGTGGAGACAGAAG | 720 (Wild-type) | Genotyping |
| YJ12530 | CCTACCTGCCTTTCCCGTTG | 420 (*Map3k13^KO/KO*) | |
| YJ12531 | TGCTCACCTGGGATCTGACTA | | |
| YJ12532 | TTAATACGACTCACTATAGGG TGGCACTACAGGTCACATAC | 121 | sgRNA2 PCR |
| YJ12535 | AAAAGCACCGACTCGGTGCC | | |
| YJ12533 | TTAATACGACTCACTATAGG GGACCTCGTACAGCTGTCCGT | 121 | sgRNA5 PCR |
| YJ12535 | AAAAGCACCGACTCGGTGCC | | |
| YJ12534 | TTAATACGACTCACTATAGGG ACTCCAGTATAGCCTCGATG | 121 | sgRNA12 PCR |
| YJ12535 | AAAAGCACCGACTCGGTGCC | | |
| YJ12536 | GGCCCAGGCCCATTATTGTG | 83 (*Map2k7-L*) | qRT-PCR |
| YJ12537 | GTTGGCCAGTGGGAGTTGCAG | | |
| YJ12538 | GCCAAGCTGAAGCAGGAGAACC | 100 (*Map2k7-S*) | qRT-PCR |
| YJ12539 | CCAGTGGGAGTTGCAGGGTG | | |

## Image acquisition and analysis

The slides of H and E staining were scanned with Nanozoomer 2.0-HT digital slide scanner (Hamamatsu Photonics) in brightfield at 40X magnification. The images were processed using NDP.view2 Viewing software (Hamamatsu Photonics, RRID: SCR_017105) and ImageJ software (ImageJ, RRID: SCR_003070). Fluorescence images of paired WT and mutant samples were acquired on a Zeiss LSM 710 confocal microscope at 40X magnification. The images were taken as Z-stack under identical settings, and the maximum intensity projection images (unless otherwise specified) were processed using ImageJ software (ImageJ, RRID: SCR_003070). For image quantification, three midline parasagittal sections per brain and at least three brains per genotype of given age were analyzed and data was averaged. Cells were counted using the cell counter plugin for ImageJ (ImageJ, RRID: SCR_003070). Analyses of cell numbers for Calbindin[+] PCs, p-c-Jun[+] PCs, dendrite swelling[+] PCs or cleaved caspase-3[+] PCs were performed by counting the soma of each PC in the entire lobules. The thickness of the molecular layer visualized by Calbindin staining was assessed for lobule V/VI in midline sections by measuring the perpendicular distance from the molecular layer-facing edge of a Purkinje cell soma to the outer edge of the molecular layer. Cerebellum area was calculated by outlining the perimeter of the outer edges of the sagittal sections of cerebellum. TUNEL[+] cells were counted by analyzing particles after adjustment of threshold and watershed. The particles with area larger than 8 $\mu m^2$ were measured. TUNEL[+] cell density was calculated the number of TUNEL[+] cells

in entire cerebellum divided by entire cerebellum area. GFAP or IBA1 immunofluorescence intensity density was calculated by dividing the GFAP or IBA1 immunofluorescence intensity of entire cerebellum by the entire cerebellum area. For p-c-Jun immunofluorescence intensity quantification, about 30 sampling area surrounding a single p-c-Jun$^+$ nucleus (ROI being 347.543 μm$^2$) per section were measured. For LZK immunofluorescence intensity quantification, about 30 sampling area surrounding a single soma of Purkinje cells (ROI being 352.943 μm$^2$) were measured in the region of interest per section. Integrated density was averaged after subtraction of background signal and adjustment of threshold.

## Immunoprecipitation and western blotting

Dissected cerebella from mice of indicated age were homogenized in an appropriate volume of cell lysis buffer (50 mM Tris.Cl [pH 7.4], 1% Triton X-100, 0.1% SDS, 1 mM EDTA [pH 7.0], 150 mM NaCl, 1% n-Octyl β-D-glucopyranoside, 1 x protease inhibitor cocktail [Roche, 05892970001]) using TissueRuptor II (QIAGEN, 9002756), then lysed for 1 hr on ice, and cleared by centrifugation at 13,000 rpm for 10 min at 4°C. Supernatants were collected and protein concentrations were determined by Pierce BCA protein assay kit (Thermo Fisher Scientific, 23225). For immunoprecipitation experiments, antibody-bound beads were prepared using 2 μg rabbit anti-MAP3K13 polyclonal antibody (Sigma-Aldrich, HPA016497) in 800 μl of lysis buffer with 50 μl of 50% Protein G agarose bead slurry, with gentle rotation at 4°C for 1 hr. ~1 mg protein lysates were pre-cleared with 50 μl of 50% Protein G agarose bead slurry, then incubated with the antibody-bound beads overnight at 4°C. The beads were washed three times with lysis buffer, and then resuspended in 60 μl lysis buffer and 20 μl 4 x Laemmli Sample Buffer (Bio-RAD, 161–0747), heat shocked in a thermomixer (Eppendorf) at 95°C for 10 min and analyzed by western blotting. Immunoprecipitated samples were separated by SDS-PAGE using Any kD Mini-PROTEAN TGX Precast Protein Gels (Bio-Rad, 4569034), and then blotted to a PVDF membrane (0.2 μm, Bio-RAD, 1620177) by Mini Trans-Blot Cell (Bio-RAD, 170–3930) at 100 mA for 1 hr. Blots were blocked in 10% non-fat dry milk in PBST (PBS with 0.05% Tween-20) for 1 hr at room temperature, and then incubated with an appropriate concentration of primary antibody in 1% non-fat BSA in PBST at 4°C for overnight. Afterwards, the membrane was incubated with Horseradish Peroxidase (HRP)-conjugated secondary antibodies (Key Resources Table) in 1% non-fat BSA in PBST at room temperature for 1 hr, followed by detection using enhanced chemiluminescence (ECL) reagents (GE Healthcare, RPN2106).

## RNA extraction, RT-PCR, and qRT-PCR

Total RNA from mouse cerebellum was extracted using TRIzol (Invitrogen, 15596018) following the manufacturers' protocols. First strand cDNA was reverse-transcribed using SuperScript IV (Invitrogen, 18091050). qPCR was run on Bio-Rad CFX96 Touch Real-Time PCR Detection System with iQ SYBR Supermix (Bio-Rad, 170–8882). Data were analyzed using CFX manager (CFX Manager, RRID: SCR_017251).

## Animal behavior tests

Mouse behavioral analysis was scored in a genotype blind manner following the protocol described in *Guyenet et al., 2010*. Briefly, ledge walking, hind limb clasping, gait, and kyphosis were scored with a scale of 0–3 in each category, resulting in total score of 0–12 points for all four measures at P30, P45, P60, P75, P90, P105, and P120. A score of 0 represents absence of the relevant phenotype and three represents the most severe phenotype. Each test was performed three times to ensure reproducibility. For data analysis, the score was calculated for each measure by taking the mean of the three measurements in each mouse.

## Statistics

GraphPad Prism 6.0 (GraphPad Prism, RRID: SCR_002798) was used for all statistical analysis. After assessing for normal distribution, statistical analyses between two groups were calculated with the two-tailed t-test for normally distributed data. For comparison of more than two groups, normally distributed data was calculated with a one-way ANOVA. Asterisks indicate significance with (*) $p < 0.05$, (**) $p < 0.01$, (***) $p < 0.001$, (****) $p < 0.0001$ for all data sets. Graphs show mean values ± standard error of the mean (SEM).

## Acknowledgements

We are grateful to our lab members for valuable discussions. We thank Drs. A D Chisholm, S L Ackerman and G Thomas for comments on the manuscript. We thank Dr. L Holzman (U Penn) for providing conditional deletion of DLK mice, UCSD Transgenic and Knockout Mouse Core for generating *Map3k13*$^{KO}$ mice, and UCSD Neuroscience Microscopy Shared Facility (NS047101) for providing imaging support. We thank A Moore, E Xu, D Arakelyan and R Zarei for technical assistance. This work was supported by funds from Howard Hughes Medical Institute, the Craig H Neilsen Foundation, the Junior Seau Foundation, and the Kavli Institute of Brain and Mind at UCSD.

## Additional information

### Funding

| Funder | Author |
| --- | --- |
| Howard Hughes Medical Institute | Yishi Jin |
| Craig H. Neilsen Foundation | Yishi Jin |
| Kavli Institute for Brain and Mind, University of California San Diego | Yishi Jin |
| Junior Seau Foundation | Yishi Jin |

The funders had no role in study design, data collection and interpretation, or the decision to submit the work for publication.

### Author contributions

Yunbo Li, Conceptualization, Data curation, Formal analysis, Investigation, Methodology, Writing - original draft, Writing - review and editing; Erin M Ritchie, Data curation, Formal analysis, Investigation, Methodology, Writing - review and editing; Christopher L Steinke, Formal analysis, Investigation, Methodology; Cai Qi, Formal analysis, Investigation, Methodology, Writing - review and editing; Lizhen Chen, Investigation, Writing - review and editing; Binhai Zheng, Resources, Supervision, Writing - review and editing; Yishi Jin, Conceptualization, Supervision, Funding acquisition, Methodology, Writing - original draft, Project administration, Writing - review and editing

### Author ORCIDs

Yunbo Li (iD) https://orcid.org/0000-0003-3222-858X
Christopher L Steinke (iD) http://orcid.org/0000-0002-1663-9971
Yishi Jin (iD) https://orcid.org/0000-0002-9371-9860

### Decision letter and Author response

Decision letter https://doi.org/10.7554/eLife.63509.sa1
Author response https://doi.org/10.7554/eLife.63509.sa2

## Additional files

### Supplementary files

• Transparent reporting form

### Data availability

This study does not generate sequencing data, proteomic data, or diffraction data. Source data for immunofluorescence quantification, cell counts, and animal behaviors have been provided for Figures 1-7.

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
