## [Decision Letter]

**Acceptance summary:**

In this very nicely revised submission an extensive amount of data are provided supporting the supporting the conclusions drawn regarding the roles of DLK and LZK in the cerebellum. Strengths include use of various Cre-inducible knockout and transgenic mouse lines of DLK and LZK and genetics to compare the role of DLK and LZK in Purkinje cells. Novel findings include that DLK and LZK are dispensable for proper cerebellar Purkinje cell development, and that increased DLK or LZK activity is sufficient to drive Purkinje cell death.

**Decision letter after peer review:**

Thank you for submitting your article "Activation of MAP3K DLK and LZK in Purkinje Cells Causes Rapid and Slow Degeneration Depending on Signaling Strength" for consideration by *eLife*. Your article has been reviewed by three peer reviewers, one of whom is a member of our Board of Reviewing Editors, and the evaluation has been overseen by Jonathan Cooper as the Senior Editor. The following individual involved in review of your submission has agreed to reveal their identity: Claire E Le Pichon (Reviewer #3).

The reviewers have discussed the reviews with one another and the Reviewing Editor has drafted this decision to help you prepare a revised submission.

Summary:

Overall, reviewers found the work in this study to be for the most part well done with a solid set of experiments with the data supporting the conclusions drawn. Strengths include use of various Cre-inducible knockout and transgenic mouse lines of DLK and LZK and genetics to compare the role of DLK and LZK in Purkinje cells. Novel findings include that DLK and LZK are dispensable for proper cerebellar Purkinje cell development, and that increased DLK or LZK activity is sufficient to drive Purkinje cell death. For publication of this manuscript in *eLife*, the following points need to be addressed.

Essential Revisions:

1) The authors need to acknowledge and discuss the fact that the Parvalbumin-Cre driver is widely expressed throughout the nervous system. This is an important point for interpretation of the experiments in this paper. While conclusions regarding Purkinje cell-related phenotypes are appropriate, interpretation of phenotypes such as behavior or early mortality is complex.

Specifically, in the DLK KO experiment, a Parvalbumin-Cre line is crossed to a Dlk flox line to delete DLK in Purkinje neurons, but also deletes DLK in many other neurons that express Parvalbumin during their development. In the over-expression models, the extreme phenotypes observed for body weight and motor behavior phenotypes are unlikely the consequence of Purkinje cell death alone, and this should be discussed.

2) For Figure 3, can the authors add an example image or close-up to illustrate the quantification in 3F of caspase- and caspase+ dendrite swellings? Please add an explanation of how a dendrite is classified as swollen or not.

3) Rostral versus caudal PV-Cre; TdT differences in expression in the cerebellum for the LZK OE are interesting; did a similar phenomenon not affect the DLK OE cross? Please comment. If not, does the uneven expression of the LZK OE transgene explain why its phenotype is less severe than the DLK OE? (one can imagine the patchy expression pattern affecting other PV+ neurons in a similar way, so if it was more homogeneous in the DLK OE mouse, this could explain the stronger phenotype and the early death etc).

4) For the over-expression models, show an image of p-cJun in rest of the brain? This can help support why the animals fail to thrive, and that the transgene effect is not restricted to the cerebellum.

---

## [Author Response]

Essential Revisions:1) The authors need to acknowledge and discuss the fact that the Parvalbumin-Cre driver is widely expressed throughout the nervous system. This is an important point for interpretation of the experiments in this paper. While conclusions regarding Purkinje cell-related phenotypes are appropriate, interpretation of phenotypes such as behavior or early mortality is complex.Specifically, in the DLK KO experiment, a Parvalbumin-Cre line is crossed to a Dlk flox line to delete DLK in Purkinje neurons, but also deletes DLK in many other neurons that express Parvalbumin during their development. In the over-expression models, the extreme phenotypes observed for body weight and motor behavior phenotypes are unlikely the consequence of Purkinje cell death alone, and this should be discussed.

We completely agree with the reviewers’ suggestion.

In the original manuscript, we stated in the first paragraph of the Discussion that “As Purkinje cells and cerebellum are not essential for animal viability, we interpret that the lethality of *Pvalb^Cre/+^*;DLK(iOE)/+ pups is likely due to disruption of other parvalbumin-expression neurons, with the underlying basis remaining to be addressed in future studies”.

In the revised manuscript, we have expanded the Discussion to emphasize that the extreme phenotypes observed for body weight and motor behavior phenotypes in the over-expression mice are unlikely the consequence of Purkinje cell death alone.

“Additionally, we observed body weight loss, motor deficits and early mortality associated with *Pvalb^Cre/+^;* Hipp11-DLK(iOE*)/+* pups. […] While our data do not address the cellular and/or circuit basis of these gross phenotypes, we infer that the grossly abnormality and lethality of *Pvalb^Cre/+^;*Hipp11-DLK(iOE)*/+* pups are unlikely the consequence of Purkinje cell death alone. The underlying basis would be of interest in future studies.”

It is our hope that by sharing our observations on these mice, it will stimulate the interests of researchers with proper expertise to investigate the mechanisms at the cellular and circuit levels.

2) For Figure 3, can the authors add an example image or close-up to illustrate the quantification in 3F of caspase- and caspase+ dendrite swellings? Please add an explanation of how a dendrite is classified as swollen or not.

By Calbindin staining, we observed the dendrites of normal Purkinje cells as smooth bush-like arbors. In degenerating Purkinje cells, the dendrites frequently showed numerous varicosities along the arbors, which we annotated as dendrite swellings in Figure 3. We showed an example of swollen dendrites containing cleaved caspase-3^+^ signals in the original Figure 3C (the enlarged panels with blue outline). We have now included a new Figure 3—figure supplement 4 showing dendrite swellings that are negative for cleaved caspase-3. We made text revision to explain our classification of dendrite swellings (subsection “Elevated DLK expression disrupts dendritic cytoskeleton”).

3) Rostral versus caudal PV-Cre; TdT differences in expression in the cerebellum for the LZK OE are interesting; did a similar phenomenon not affect the DLK OE cross? Please comment. If not, does the uneven expression of the LZK OE transgene explain why its phenotype is less severe than the DLK OE? (one can imagine the patchy expression pattern affecting other PV+ neurons in a similar way, so if it was more homogeneous in the DLK OE mouse, this could explain the stronger phenotype and the early death etc).

The induced expression patterns from both DLK(iOE) and LZK(iOE) transgenes using the *Pvalb^Cre^* were similar, starting stronger in anterior lobules in the cerebellum. This is shown in the revised Figure 2—figure supplement 1D (outlined on the P10 image). The rapid death of Purkinje cells in *Pvalb^Cre/+^;*Hipp11-DLK(iOE)/+ animals makes the pattern of tdTomato expression appearing different from those for *Pvalb^Cre/+^;*Hipp11-LZK(iOE)/+ (in Figure 4E and Figure 4—figure supplement 1D). We also showed that p-c-Jun levels in Purkinje cells of *Pvalb^Cre/+^;*Hipp11-DLK(iOE)/+ mice were stronger than those in *Pvalb^Cre/+^;*Hipp11-LZK(iOE)/+ mice (see Figure 5—figure supplement 1J, K, L). Therefore, we conclude that the phenotypic differences between *Pvalb^Cre/+^;*Hipp11-LZK(iOE)/+ and *Pvalb^Cre/+^;*Hipp11-DLK(iOE)/*+* are not due to the patchy expression, rather the signaling strength.

4) For the over-expression models, show an image of p-cJun in rest of the brain? This can help support why the animals fail to thrive, and that the transgene effect is not restricted to the cerebellum.

We have included such images showing p-c-Jun in cortex of *Pvalb^Cre/+^;*Hipp11-DLK(iOE)/+ and *Pvalb^Cre/+^;*Hipp11-LZK(iOE)/+, respectively, in a new Figure 3—figure supplement 1 and a new Figure 4—figure supplement 3. We observed basal levels of p-c-Jun in cells not expressing either kinase and p-c-Jun staining in those marked by tdTomato as induced by elevated DLK or LZK expression.

As responded to point 1, we agree that the transgene effect is not restricted to the cerebellum. In this manuscript, we have focused our analysis on the effects on cerebellar Purkinje cells. We hope to dissect the underlying basis for other abnormalities in *Pvalb^Cre/+^;*Hipp11-DLK(iOE)/+ in future studies.